# Digital plasmonic nanobubble detection for rapid and ultrasensitive virus diagnostics

Yaning Liu[1,7], Haihang Ye [1,7 ✉], HoangDinh Huynh[2], Chen Xie[1], Peiyuan Kang[1], Jeffrey S. Kahn[2,3] & Zhenpeng Qin [4,5,6 ✉]

Rapid and sensitive diagnostics of infectious diseases is an urgent and unmet need as evidenced by the COVID-19 pandemic. Here, we report a strategy, based on DIgitAl plasMONic nanobubble Detection (DIAMOND), to address this need. Plasmonic nanobubbles are transient vapor bubbles generated by laser heating of plasmonic nanoparticles (NPs) and allow single-NP detection. Using gold NPs as labels and an optofluidic setup, we demonstrate that DIAMOND achieves compartment-free digital counting and works on homogeneous immunoassays without separation and amplification steps. DIAMOND allows specific detection of respiratory syncytial virus spiked in nasal swab samples and achieves a detection limit of ~100 PFU/mL (equivalent to 1 RNA copy/μL), which is competitive with digital isothermal amplification for virus detection. Therefore, DIAMOND has the advantages including one-step and single-NP detection, direct sensing of intact viruses at room temperature, and no complex liquid handling, and is a platform technology for rapid and ultrasensitive diagnostics.

[1] Department of Mechanical Engineering, University of Texas at Dallas, Richardson, TX 75080, USA. [2] Departments of Pediatrics, University of Texas Southwestern Medical Center, 5323 Harry Hines Boulevard, Dallas, TX 75390, USA. [3] Departments of Microbiology, University of Texas Southwestern Medical Center, 5323 Harry Hines Boulevard, Dallas, TX 75390, USA. [4] Department of Surgery, University of Texas Southwestern Medical Center, 5323 Harry Lines Blvd, Dallas, TX 75390, USA. [5] Department of Bioengineering, University of Texas at Dallas, Richardson, TX 75080, USA. [6] Center for Advanced Pain Studies, University of Texas at Dallas, Richardson, TX 75080, USA. [7] These authors contributed equally: Yaning Liu, Haihang Ye. ✉email: Haihang.Ye@utdallas.edu; Zhenpeng.Qin@utdallas.edu

The ability to rapidly detect diseases with high precision is of paramount importance as evidenced by the current COVID-19 pandemic[1,2]. Digital immunoassays have been a remarkable conceptual advance over the past two decades due to their capabilities of single-molecule detection and absolute quantification[3–9]. They partition the analytes into microwells or emulsion droplets as small compartments for independent signal amplification and digital counting, leading to the sensitivity enhancement by up to $10^3$-fold over the conventional immunoassays (i.e., enzyme-linked immunosorbent assay)[3,7,10]. Despite these advantages, digital immunoassays have complex assay protocols, such as multiple washing steps, sample partitioning, and signal amplification prior to the measurements, which have limited their widespread use. Such paradigms prompt further innovations to simplify the digital immunoassays. Nanoparticles (NPs) are thus employed as labels and provide an alternative to streamline the workflow. For example, interrogation of individual NPs through bright-field or dark-field imaging[11], interferometric[12] or fluorescent imaging[13,14], surface-enhanced Raman scattering[15], and surface plasmon resonance microscopy imaging[16] circumvents the signal amplification step, while tracking the Brownian motion of individual particles allows digitizing a homogeneous immunoassay without washings[17–19]. However, those techniques still rely on multistep operation, cumbersome chip preparation, tag labeling, and advanced imaging systems and are currently confined to laboratory settings.

Here, we report a strategy for a simplified digital immunoassay based on DIgitAl plasMONic nanobubble Detection (DIAMOND, see Table S1 for comparison with other assays). Plasmonic nanobubbles (PNBs) refer to the vapor bubbles generated by short laser pulse excitation of plasmonic NPs (e.g., gold NPs or AuNPs) and amplify their intrinsic absorption for the detection by a secondary probe laser[20–24]. PNBs have lifetimes that last nanoseconds and are sensitive to physical properties of NPs such as size, shape, concentration, and clustering state[20–22]. Taking advantage of these unique properties, we designed an optofluidic setup to flow the NP suspension in a micro-capillary, where two aligned laser beams synchronically activate the NPs and detect the PNBs (Figs. 1a and S1). Since the PNBs are transient events and have no crosstalk between laser pulses, DIAMOND creates microscale "virtual detection zones" of about 16 pL and counts the "on" and "off" signals in a compartment-free manner (Fig. 1b, c). Although its detection scheme is similar to flow cytometry, DIAMOND utilizes non-fluorescent labels for signal generation and detects analytes without flow focusing[25,26]. Using AuNPs and silica ($SiO_2$) beads as models, we demonstrate that DIAMOND detects a single NP and can be implemented on a homogenous assay with a sub-femtomolar detection limit. When applied to detect the respiratory syncytial virus (RSV), DIAMOND provides good detection specificity among closely related respiratory viruses and achieves a detection limit of ~100 PFU/mL or 1 RNA copy/μL equivalent. With rapid readouts and ultrasensitive detection, DIAMOND opens possibilities to develop digital immunoassays without the need for separation, amplification, and physical compartments (such as droplets or microwells) at room temperature.

## Results
### DIAMOND allows single-nanoparticle (NP) detection and sizing.
We first evaluated the ability of DIAMOND for single-NP detection. Suspensions of 75 nm AuNPs (as characterized in Figs. S2, S3, and Table S2) were injected into a 200 μm capillary by a syringe pump with a flow rate of 6 μL/min (Fig. S1a). The fluid was irradiated and detected synchronously by the aligned pump and probe lasers (Fig. S1b and S1c). We set a laser scanning

speed of 1000 μm/s and flow speed of 2500 μm/s to avoid particles being counted twice. For simplicity, we converted the particle concentration into the expected average number ($\lambda$) of AuNPs per detection zone:

$$\lambda = c \cdot V \qquad (1)$$

where $c$ is particle concentration (NPs·mL$^{-1}$) and $V$ is 16 pL. Figure 2a shows the representative DIAMOND results for the given $\lambda$, where discrete PNB signals with fluctuated intensity could be observed when $\lambda \leq 4$. To explain this, we plotted the Poisson distributions (Fig. S4) based on the below equation:

$$f(k\,;\lambda) = P(x = k) = \frac{\lambda^k e^{-\lambda}}{k!} \qquad (2)$$

where the $k$ is the actual AuNP number per detection zone, and $P$ is the Poisson probability. It is evident that the probability of zero AuNP ($k = 0$) passing through the detection zone increases with decreasing $\lambda$ and thus leads to discrete PNB signals. In particular, in the case of $\lambda = 0.04$, each PNB signal highlighted in Fig. 2a refers to a single AuNP being detected according to the Poisson distribution (Fig. S4d). This result confirms the capability of DIAMOND for single-NP detection. To correlate the PNB signals with AuNP concentration, we calculated the frequencies of positive PNB signals in each trace as "on" signals ($f_{on}$) and plotted it as a function of $\lambda$ (Fig. 2b). A high correlation coefficient ($R^2 = 0.998$) indicates the accurate AuNP quantification via $f_{on}$ counting. Furthermore, we found a linear relationship (slope = 0.985, $R^2 = 0.998$, Fig. 2c) between the theoretical probability ($P$) predicted by Poisson statistics and the experimental $f_{on}$, suggesting that DIAMOND is a calibration-free technique and allows absolute quantification.

Next, we used DIAMOND to detect AuNPs of different sizes. AuNP suspensions (15, 35, 50, and 75 nm) at the same concentration (Figs. S2, S3, and Table S2) were prepared for the test. Figure 2d shows the representative PNB signal traces. Due to a large number of AuNPs per detection zone ($\lambda = 240$), consistent PNB signals were observed in each laser pulse. Figure 2e shows representative PNB signals taken from each case and suggests that larger NP size leads to larger PNB signals. To quantify this relationship, we extracted the values of amplitude and area-under-the-curve (AUC) from each PNB signal (Fig. S5) and plotted their distribution profiles as a function of AuNP sizes (Fig. 2f). They could be fitted with 1.98- and 3.51-order dependences on the NPs' size, respectively. This seems to agree with the fact that the PNB generation is related to the amount of heat generation and absorption cross-section of AuNPs[27,28]. The good fitting indicates the PNB is a size-dependent event, and its signal can indicate the NP size.

### DIAMOND allows characterizing heterogenous NP suspensions.
We then evaluated the ability of DIAMOND to detect a heterogeneous population of NPs. The rationale is that the NP clusters or aggregates discussed later may act as bigger particles. We first tested detecting 75 nm AuNPs in the background of 15 nm NPs. Figure 3a shows the representative PNB signal traces taken from three sets of samples, including 15 and 75 nm AuNPs and a mixture of them. The large intensity variation of PNB signals observed in the mixture sample is due to the addition of 75 nm AuNPs. To sort those specific signals resulting from 75 nm AuNPs in the mixture, we developed a three-step analytical protocol. First, we extracted the amplitude and AUC values of each PNB signal as indices for quantification and plotted the corresponding results in bivariate (Fig. 3b). Individual scatters represent independent PNB events measurements. Second, we fitted normal distributions to the amplitude and AUC of 15 nm

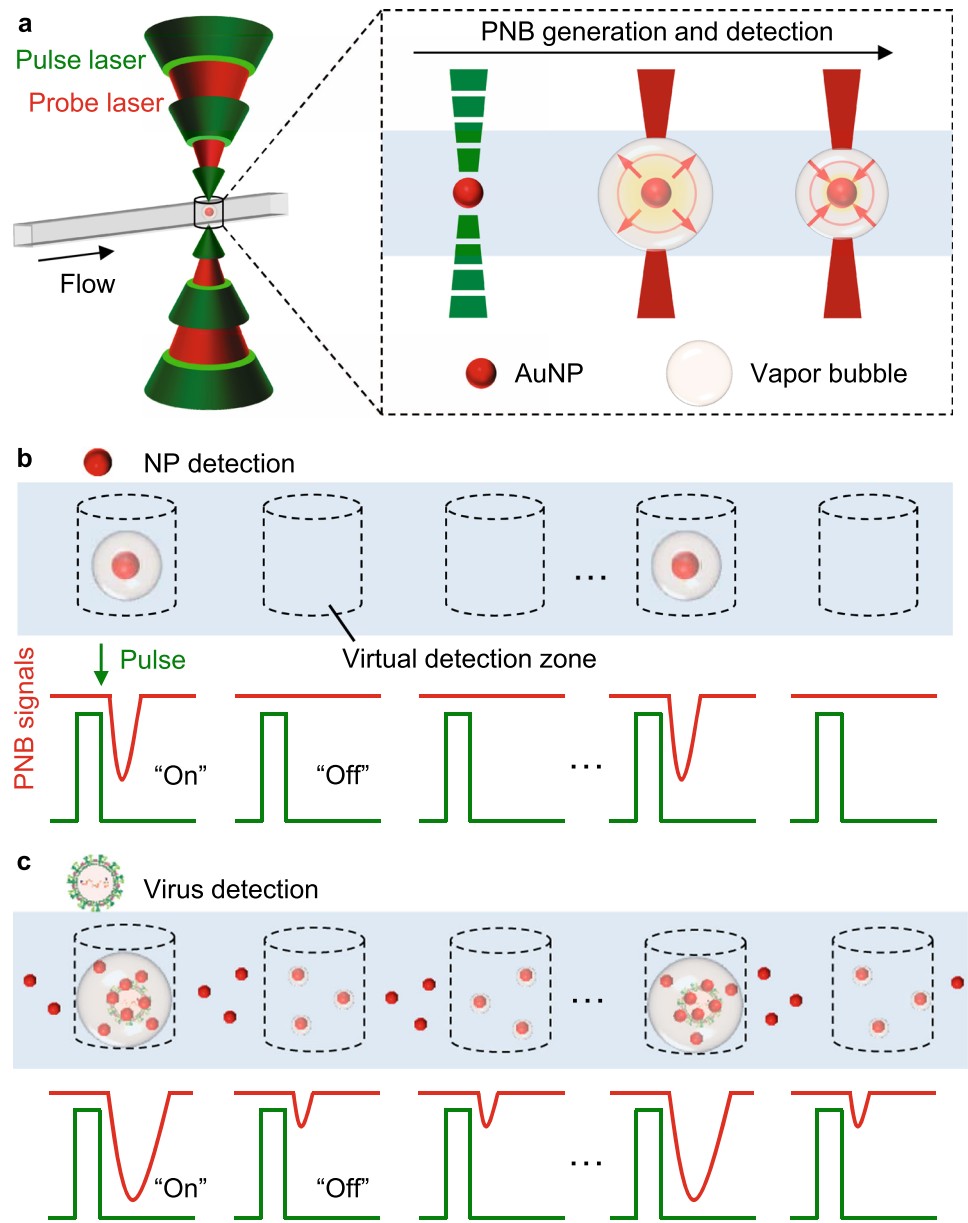

**Fig. 1 Concept of DIgitAl plasMONic nanobubble Detection (DIAMOND). a** The schematic illustration of the plasmonic nanobubbles (PNBs) generation and detection in flow. The gold nanoparticles (AuNPs) as labels are used for the generation of the PNBs by short laser pulses and subsequently detected by a secondary probe laser due to the amplified optical absorption. **b**, **c** The principle of digital counting for NP (**b**) and virus (**c**) detection. The "on" and "off" in (**b**) and (**c**) refer to the positive and negative PNB signals representing for the presence or absence of targets (i.e., NP and virus, respectively). The laser pulses create "virtual" detection zones as compartmentations for independent counting of PNB signals. The three dots shown in (**b**) and (**c**) refer to ellipses and indicate the omission of detection events in a repeated operation.

AuNPs as a reference (Fig. 3b, case i) and calculated the threshold ($T$) using five times standard deviations ($\sigma$) above the mean ($\mu$). As such, the two thresholds covered over 99.9% of the scatters. Last, we assigned the thresholds to the mixture sample for data sorting. The scatters above the thresholds refer to the positive or "on" PNB signals resulting from 75 nm AuNPs in the mixture (Fig. 3b, case ii). For direct comparison, we benchmarked it with $f_{on}$ from 75 nm AuNPs alone (thresholds = 0, Fig. 3b, case iii) and the Poisson probability (Fig. 3c). The agreement between experimental results and theoretical prediction indicates that our method provides accurate and absolute quantification of large NPs from a strong background of small ones (i.e., 1 in 240). Importantly, the capability of DIAMOND to detect heterogeneity without an additional separation step may have

potential implications as an analytical tool for NPs and requires further study.

**DIAMOND allows sensitivity enhancement for a homogeneous assay using NPs.** We further evaluated the feasibility of implementing DIAMOND for the homogeneous assay. Homogeneous assays are simple, one-step sensing methods that require minimal liquid handling and are promising for rapid detection[29]. To validate our hypothesis in a model system, we used the AuNPs as probes to detect $SiO_2$ beads (Fig. 4a). Figure 4b shows the transmission electron microscopy (TEM) image of the as-prepared product, where a core-satellites structure was formed with AuNPs fully covering the surface of $SiO_2$ beads. Figure 4c, d

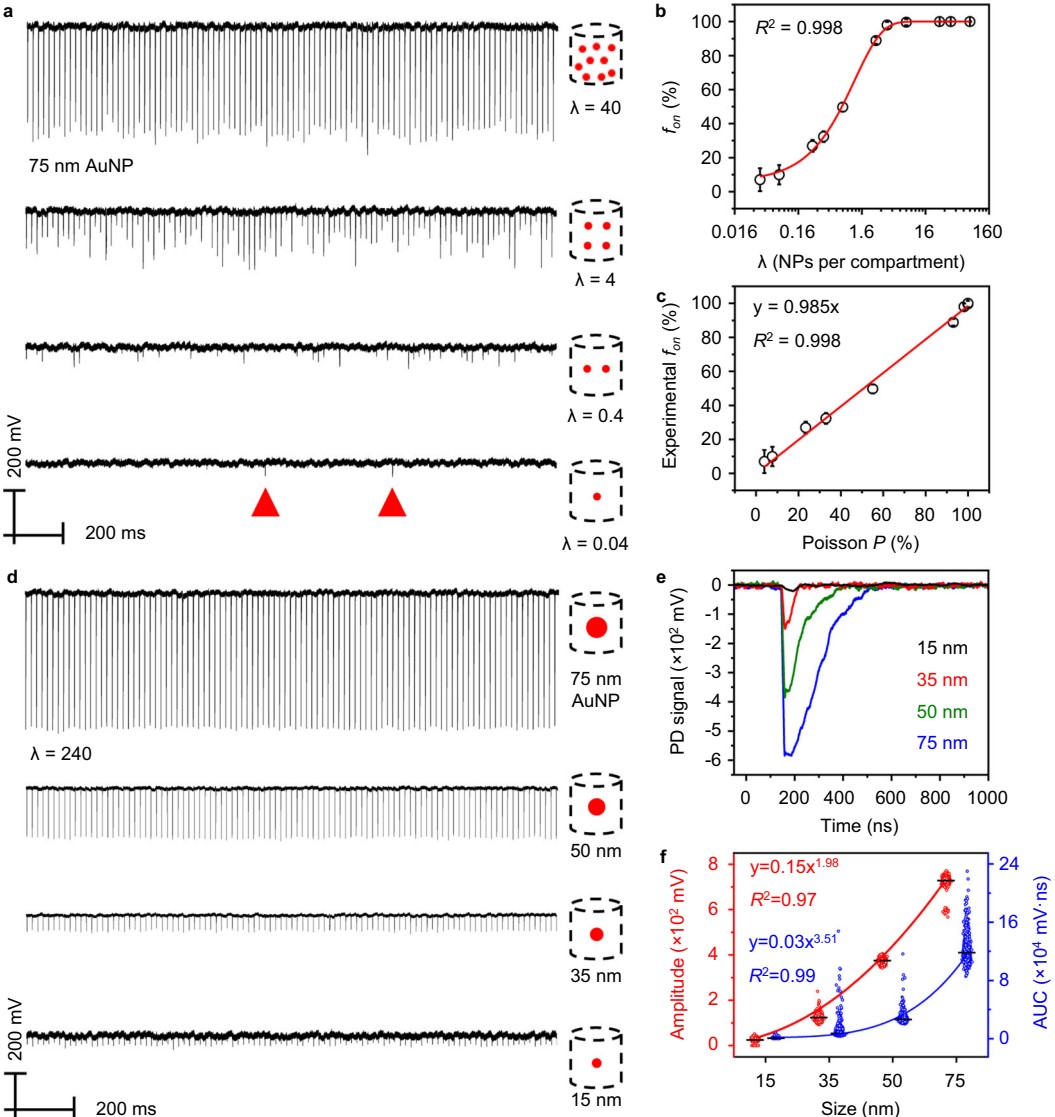

**Fig. 2 Detection of AuNPs and size differentiation by DIAMOND. a** Representative PNB signal traces (100 pulses) for 75 nm AuNP suspensions with different particle concentrations. Schematics represent the decreased number ($\lambda$) of AuNPs per detection. Red triangles mark the positive PNB signals. **b** Quantification of AuNP concentrations by plotting the frequency count ($f_{on}$) against $\lambda$. A one-phase exponential association fitting is applied for the calibration curve. **c** Linear correlation between experimental $f_{on}$ and the theoretical probability ($P$) based on Poisson statistics. **d** Representative PNB signals traces (100 pulses) for AuNP suspensions with different sizes. Schematics represent the increasing AuNP sizes and same AuNP number per detection. **e** Individual PNB signals extracted from (**d**). PD is photodetector. **f** Correlations between the amplitude and area-under-curve (AUC) of PNB signals as a function of AuNP size. Black lines denote the statistical average values. Error bars in (**b**) and (**c**) indicate the standard deviations of three independent measurements.

shows representative PNB signal traces and corresponding bivariate plot taken from serial assay solutions, respectively. Similarly, the thresholds ($T = \mu + 5\sigma$) of amplitude and AUC were calculated from the control group (highlighted in dashed lines) and assigned for counting "on" signals. The $f_{on}$ was then plotted as a function of $\lambda$ or particle concentration of $SiO_2$ beads (Fig. 4e). A linear relationship ($R^2 = 0.99$) in the range of 0.0016–0.16 was observed and the limit of detection (LOD) was calculated to be 0.0028, equivalent to $1.75 \times 10^5 \, mL^{-1}$ or 290 aM (inset of Fig. 4e). Such detection limit is ~570-fold lower than the colorimetric detection (Fig. S6). It should be pointed out that the detection range of DIAMOND only covers 2 logs due to the limited counting number (i.e., 3000 pulses). Alternatively, we can use an analog method to analyze PNB signals (e.g., averaged AUC versus analyte concentration), which should provide an additional detection range beyond 100% $f_{on}$[12,30]. Furthermore,

DIAMOND allows absolute quantification for homogeneous assays, as indicated by the linear correlation (slope = 1.007, $R^2 = 0.999$) between the background-subtracted frequency ($f_{on}'$) and the Poisson probability (Fig. 4f).

**DIAMOND allows rapid and sensitive diagnosis of intact virus.** Finally, we applied DIAMOND to rapidly detect the respiratory syncytial virus (RSV). RSV is the major respiratory pathogen that accounts for up to 74,500 deaths in 2015 globally in children below age 5[31]. We chose Synagis (Palivizumab) as the detection antibody and conjugated it on the AuNPs through 3,3′-dithiobis (sulfosuccinimidyl propionate) as a cross-linker (Fig. S7). We used 15 nm AuNPs as labels because they can be prepared with uniform size and shape. The AuNP-Synagis probes specifically recognized the fusion proteins on the RSV surface at room

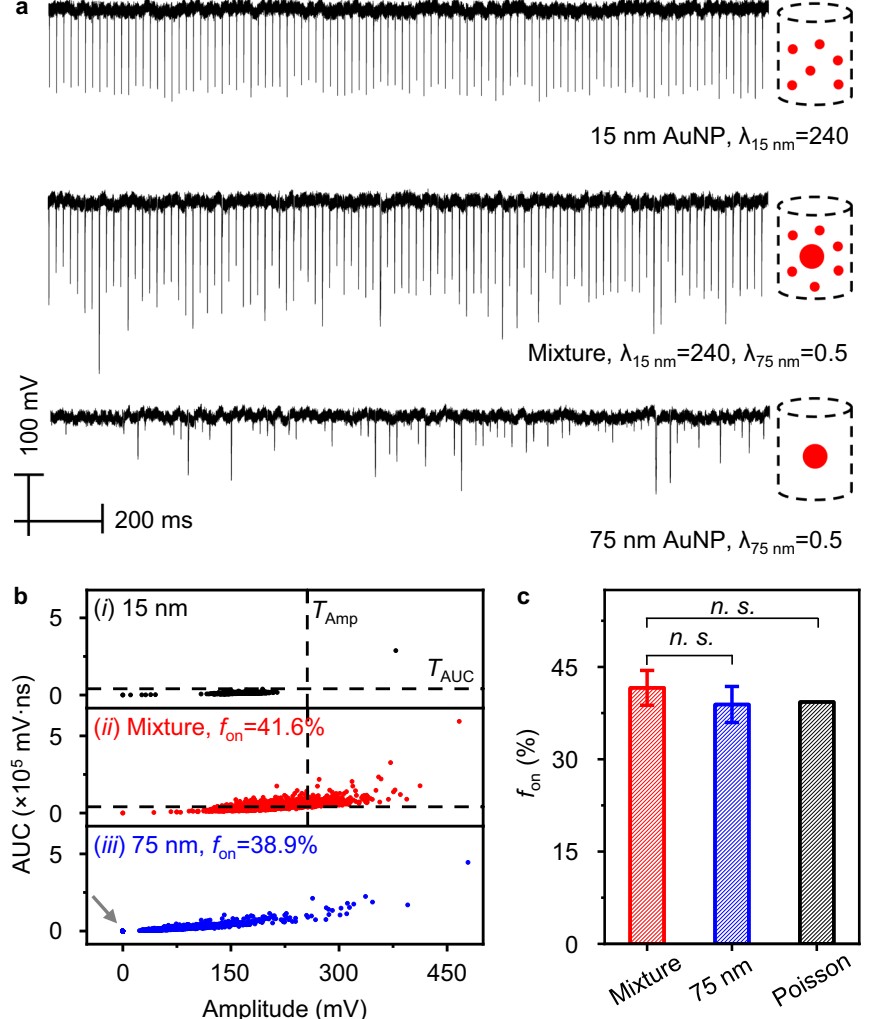

**Fig. 3 Identification of large AuNP in a heterogenous AuNP solution by DIAMOND. a** Representative PNB signals traces (100 pulses) for 15 and 75 nm AuNPs and their mixture. Schematics represent the sample information. **b** Bivariate plots of amplitude and AUC extracted from 3000 pulses for the three samples in (**a**). Cases (i–iii) refer to the suspensions of 15 nm, mixture, and 75 nm AuNPs, respectively. The dashed lines in cases (i) and (ii) indicate the same thresholds of amplitude ($T_{Amp}$) and AUC ($T_{AUC}$). The gray arrow in case (iii) highlights scatters at 0. **c** Bar plot of experimental frequencies ($f_{on}$) as determined in (**b**) for cases (ii) and (iii) and theoretical probability predicted by Poisson statistics. n.s. stands for no significant difference ($p$-value > 0.05). The error bars for experimental $f_{on}$ indicate the standard deviations of three independent measurements.

temperature and were ready for detection (Fig. 5a). Figure 5b shows the TEM images of AuNP probes targeting the RSV of different morphologies (highlighted in red). Figures S8 and 5c show the colorimetric detection of different respiratory viruses, including Parainfluenza viruses (PIV), Influenza viruses A (IVA), Human metapneumovirus (hMPV), and purified RSV. The results suggest the good detection specificity using AuNP probes and a LOD of $3.6 \times 10^4$ PFU/mL for RSV in colorimetric detection. For a direct comparison, we benchmarked the immunoassay results against the commercially available lateral flow immunoassay (LFIA) kit (BinaxNOW, Abbott), which has a LOD of $1.6 \times 10^4$ PFU/mL (Fig. S9). We then carried out DIAMOND for RSV detection and provided representative detection results in Fig. S10. Following the same protocol discussed above, we set the thresholds based on the control group (Fig. 5d) and assigned them for $f_{on}$ counting (Fig. 5e). In particular, a linear relationship ($R^2 = 0.995$) in the range of $10^2$–$10^4$ PFU/mL was observed, and the LOD was calculated to be 108 PFU/mL (inset of Fig. 5e). Evidently, DIAMOND achieved a sensitivity enhancement of ~333-fold over the colorimetric result of homogeneous immunoassay and ~150-fold over the commercial LFIA.

To further demonstrate the potential clinical applications, we applied DIAMOND to detect RSV spiked in the nasal swab samples. It should be mentioned that when used for control virus detection (i.e., hMPV, PIV, and IVA, dispersed in borate buffer), the RSV-specific AuNP probes caused non-specific binding as suggested by the positive PNB signals (Figs. S11 and 6a, b). This phenomenon can be ascribed to that the control viruses were received in cell culture fluid that contains impurities like cell debris and thus leads to the non-specific aggregation. To address this issue, we used bovine serum albumin (BSA) to backfill the AuNP probes (inset of Figs. 6c and S12). The BSA has been frequently used in immunoassays as block reagent and can prevent the non-specific binding for improved detection specificity[32]. To evaluate the performance of BSA-backfilled AuNP probes, we incubated them with respiratory viruses spiked in nasal swab samples and subjected the complete assay solution to the DIAMOND test. The viral transport medium (VTM), hMPV, PIV, and IVA were used as negative controls, and all viral titers were kept the same as $10^5$ PFU/mL. Figures S13 and 6c, d show the virus detection results using BSA-backfilled probes, where the positive PNB signals from the control viruses were

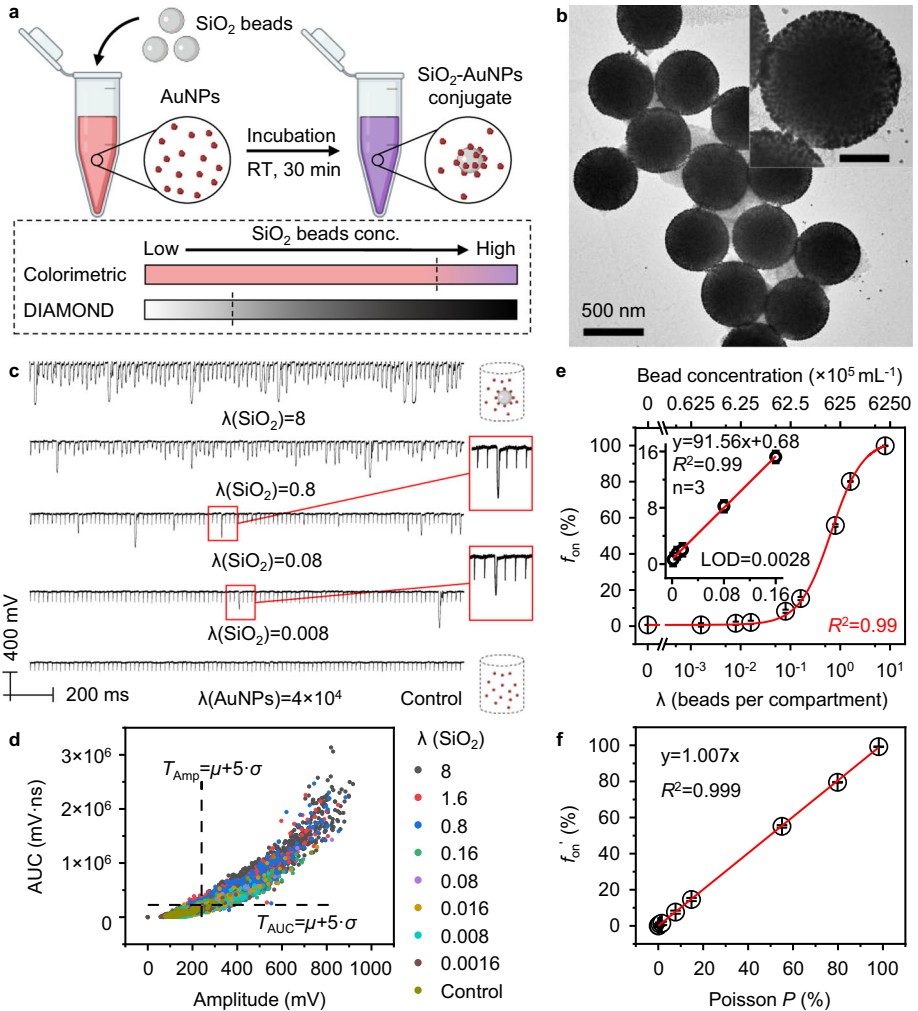

**Fig. 4 Detection of SiO$_2$ beads in a homogeneous assay by DIAMOND. a** Schematic of a homogeneous assay of SiO$_2$ beads by AuNPs as a pair of targets and probes at room temperature (RT). Lower panel shows that when bead concentrations are insufficient to induce the color change, DIAMOND can detect the PNB signals. **b** TEM image of SiO$_2$-AuNPs conjugates. Scale bar in inset is 200 nm. **c** Representative PNB signal traces (100 pulses) for the assay solutions. Schematics represent the assay information. **d** Bivariate plot of amplitude and AUC extracted from 3000 pulses for the assay solutions with different λ of SiO$_2$ beads. Dashed lines indicate the positions of thresholds calculated from the control sample. **e** Quantification of SiO$_2$ bead concentration and λ(SiO$_2$) as a function of frequency counting ($f_{on}$). A logistic fitting is applied for the calibration curve. Inset shows the linear region of the calibration curve. The LOD was calculated as 3 standard deviations of the control dividing the slope of regression line. **f** Linear correlation between the background-subtracted frequency ($f_{on}'$) and Poisson probability. Error bars in (**e**) and (**f**) indicate the standard deviations of three independent measurements.

significantly reduced, while the PNB signals from RSV can be easily distinguished from the control samples, yielding $f_{on} = 100\%$ that matches well with the theoretical probability. In contrast, the non-backfilled probes still cause non-specific aggregation upon detection (Fig. S14). This result suggests that DIAMOND has a better detection specificity when utilizing BSA-backfilled probes. Similar to that of the purified viruses (Fig. 5), these probes enable sensitive detection of spiked RSV in nasal swab samples with a detection limit of 102 PFU/mL (Figs. S15 and 6e, f). Taken together, these data demonstrated that implementing DIAMOND on a homogeneous immunoassay allows sensitive analysis of viral samples in the human specimen matrix and supports the potential clinical applications.

To compare DIAMOND with other state-of-the-art methods, we performed measurements using digital loop-mediated iso-thermal amplification (dLAMP). dLAMP is a rapid molecule test and provides absolute quantification of nucleic acids, which has been used to quantify a variety of viruses[33–36]. To perform the dLAMP, we used a commercially available fluorescent LAMP kit

and microwell chips to detect genomic RNA from RSV (A2 strain). A set of primers (Table S3) were used according to a previous publication[37]. Figure S16a–e shows the fluorescence images of the dLAMP results after incubating at 65 °C for 30 min, where the fraction of positive wells (brighter) reduces as the concentration of target RNA decreases. A calibration curve ($R^2 = 0.9998$, Fig. S16f) was established as a function of the RNA inputs. The LOD was estimated to be 2 copies/μL using synthetic RNA (inset of Fig. S16f). We then used this method to detect RSV from the spiked samples. The RSV extracts were collected using a commercially available RNA extraction buffer and purification kit. Figure S17a–d shows the dLAMP detection results for RSV extracts at different concentrations. Specifically, dLAMP has a LOD of ~200 PFU/mL (Fig. S17e), similar to that of DIAMOND. Furthermore, referring to the calibration curve in Fig. S16f, we estimated that 100 PFU/mL is equivalent to 1 copy/μL RNA with our protocols. In other words, DIAMOND has a LOD of single-molecule detection, which is competitive with nucleic acid amplification methods.

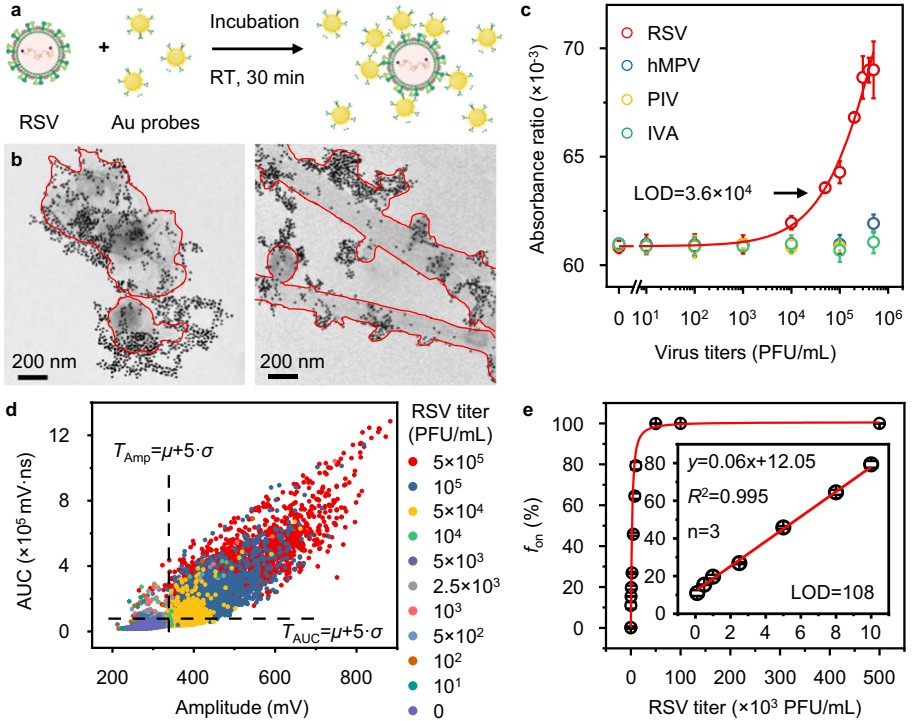

**Fig. 5 Detection of respiratory syncytial virus (RSV) in a one-step homogenous immunoassay by DIAMOND. a** The schematic illustration of a homogeneous immunoassay for RSV utilizing antibody-conjugated AuNPs as probes at room temperature (RT). **b** TEM images of AuNP probes targeting RSV. The boundaries of RSV were highlighted in red. **c** Colorimetric analysis of the AuNP-based homogeneous immunoassay for different respiratory viruses. hMPV is Human metapneumovirus, PIV is Parainfluenza viruses, and IVA is Influenza A. **d** Bivariate plot of amplitude and AUC extracted from 3000 pulses for the assay solutions with RSV at different titers (PFU/mL). Dashed lines indicate the positions of thresholds calculated from the control sample. **e** Quantification of RSV titers as a function of $f_{on}$. Inset shows the linear detection range. Error bars in (**c**) and (**e**) indicate the standard deviations of three independent measurements, the data were fitted by logistic fitting, and the LOD was calculated as 3 standard deviations of the control divided by the slope of regression line.

## Discussion

Digital immunoassays create high standards as next-generation diagnostic platforms, such as calibration-free quantification and single-molecule detection of biomarkers for early diagnostics. The major barriers to its widespread use are the time-consuming protocols and laboratory infrastructures. Here we developed DIAMOND to overcome some of these bottlenecks. Specifically, DIAMOND uses a homogeneous immunoassay format and does not require additional sample washing, separation, and signal amplification steps[4,6]. On the other hand, DIAMOND does not require chip preparation or on-chip reaction and can be performed with less assay time than the digital homogeneous assays[17–19]. Also, DIAMOND can detect intact viruses without additional liquid handling (i.e., virus extraction, thermal incubation, and chip loading), and thus offer a simplified diagnostic approach at room temperature.

In the present study, we focused on developing and validating a versatile digital immunoassay. We envision several further improvements to this technology in order to bring it to a broad range of labs and practical applications. First, it is possible to design a small benchtop device for the plasmonic nanobubble (PNB) measurements that can be distributed to other labs[38]. By replacing the research-grade picosecond pulse laser with a smaller nanosecond (ns) laser (e.g., Wedge-HB-532, RPMC), all components can be integrated into a benchtop device ($15 \times 15 \times 6$ inches, Fig. S18). Evaluation of PNB generation and detection by this ns laser shows robust results across a range of AuNP concentrations (Fig. S19). More importantly, the ns laser provides a repetition rate up to 2000 Hz for a much faster readout and thereby more efficient event counting. Second, further optimization of the optofluidic system can increase the sampling efficiency for the PNB measurements. In the present system, we used the readily available micro-capillary although the laser probes only a small portion of the sample (20–40% along and 5–10% orthogonal to the flow direction, respectively, Fig. S20). A low sampling efficiency leads to fewer events counted for a given sample volume and thus limits the dynamic range and sensitivity for the detection[5]. A microfluidic flow-focusing system can readily solve this problem by creating a narrower flow path[39]. Taking advantage of the high-throughput ns laser and focusing flow, we expect to increase the event counting (e.g., 1 million readings within 10 min, similar to flow cytometer). This will improve the detection range and sensitivity of DIAMOND (Fig. S21 and Table S4) because the digital counting performance essentially relies on the number of counted events[40]. Last, it is worth exploring different modes of DIAMOND operation. Since the PNB generation is dependent on the laser fluence, we can perform the DIAMOND test in two modes by modulating the laser fluence above and below the PNB generation threshold[20], referred to as above- and below-threshold modes. Currently, we used the above-threshold mode with a high laser fluence at $P_{PNB} = 100\%$ (blue arrow of Fig. S22), resulting in the generation of PNB signals from both the small NPs and larger clusters. Alternatively, we may adopt the below-threshold mode at a lower laser fluence ($P_{PNB} = 0\%$, red arrow of Fig. S22) to only activate large clusters for PNB generation.

It is also possible to explore the feasibility of DIAMOND to detect smaller analytes than viral particles, such as protein

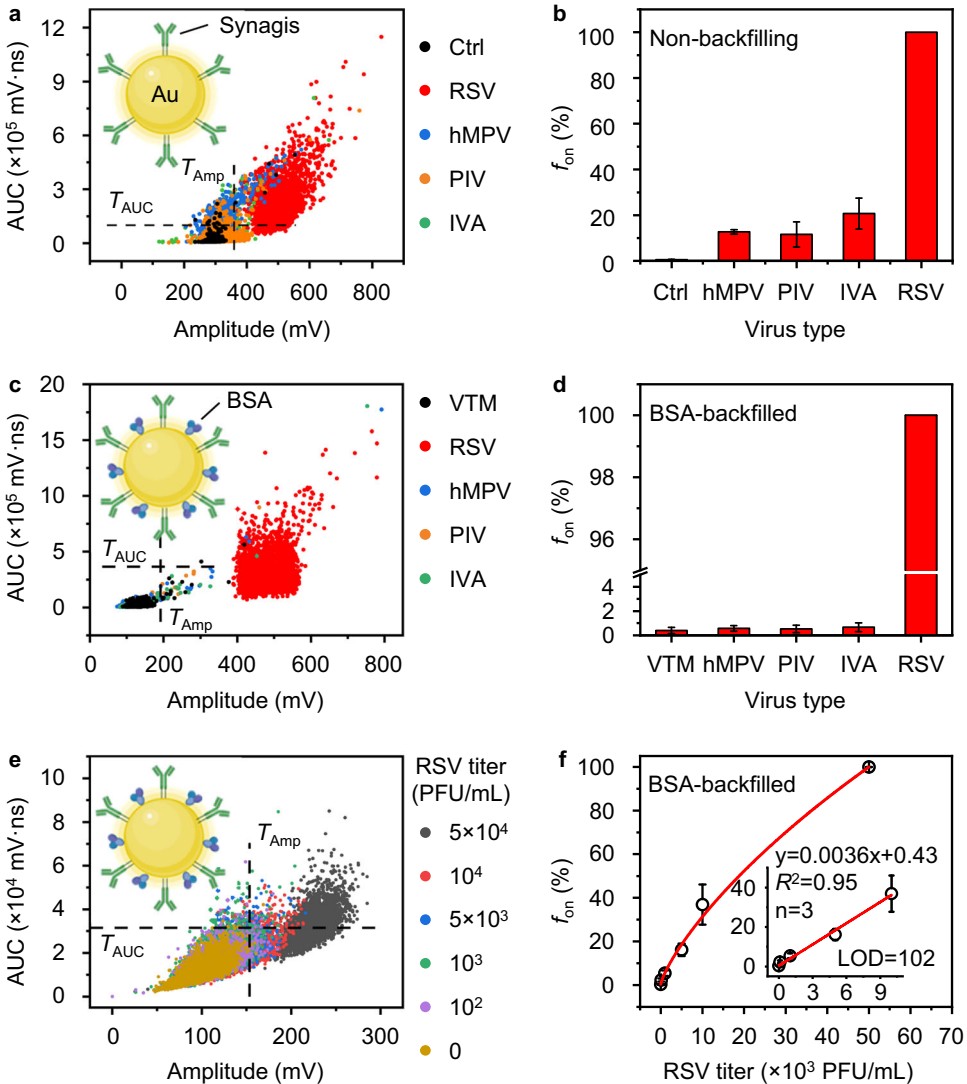

**Fig. 6 Detection of RSV among closely related respiratory viruses by DIAMOND. a, c, e** Bivariate plots of amplitude and AUC extracted from 3000 pulses for the assay solutions incubating **a** AuNP-Synagis probes with different respiratory viruses, **c** bovine serum albumin (BSA)-backfilled AuNP-Synagis probes with different respiratory viruses spiked in the nasal swab samples, and **e** BSA-AuNP probes with RSV of different titers (PFU/mL) spiked in the nasal swab samples. **b, d, f** The corresponding $f_{on}$ counted from **a, c, e** against **b, d** different respiratory viruses and **f** RSV with different titers, respectively. VTM: viral transport medium. Insets in (**a, c, e**) show the schematic of AuNP-based probes used correspondingly. Inset in (**f**) shows the linear detection range and LOD. Error bars in (**b**), (**d**), and (**f**) indicate the standard deviations of three independent measurements.

biomarkers. This, together with multiplexed detection, is important yet challenging for in vitro diagnosis[10,18,41]. Due to the small size and limited binding sites of proteins, they can only absorb a few NPs on the surface. As a result, it may be challenging to reliably differentiate protein-specific PNB signals from background signals. Therefore, we can set up a diluted setting with a low NP number per detection zone (e.g., $\lambda = 0.1$) so that the signals from zero, single, and multiple NPs passing through are unique[10]. In this case, it is crucial to ask whether DIAMOND can recognize PNB signals from events when single and multiple NPs pass through together. As a first step, we have tested and validated a threshold-based gating method[42] to determine the PNB signals generated from a specific number ($k = 0, 1, 2...$) of AuNPs (Fig. S23) using the case of $\lambda = 0.4$ shown in Fig. 2a as a model. In addition, multiplexed detection is also a critical aspect of a competitive diagnostic method. Future studies can focus on demonstrating the feasibility of multiplexing protein detection.

In summary, we have developed DIAMOND for virus detection via the AuNPs-based homogeneous immunoassay. DIAMOND

allows single-NP detection and identifies large particles from a heterogeneous population. Such capability allows DIAMOND to be implemented on homogeneous assays without sample washing. Importantly, DIAMOND can specifically detect RSV among different respiratory viruses and achieve a detection limit of ~100 PFU/mL (or 1 RNA copy/μL). Compared with other digital assays, DIAMOND counts events in a compartment-free manner and utilizes a one-pot protocol for sample handling at room temperature. Our study provides a digital counting platform for rapid and ultrasensitive diagnostics of intact viruses at their early representation.

## Methods
**Materials**. Tetrachloroauric(III) trihydrate (HAuCl$_4$·3H$_2$O, 16961-25-4, 99.9%), sodium citrate tribasic dihydrate (Na$_3$CA·2H$_2$O, 6132-04-3, ≥99%), and hydroquinone (123-31-9, ≥99%), sodium chloride (NaCl, 7647-14-5, ≥99.5%) were purchased from Sigma-aldrich. 3,3′-dithiobis (sulfosuccinimidyl propionate) (DTSSP, 21578, 50 mg) and borate buffer (1M, 28341) were purchased from Thermo Scientific. Sucrose ((57-50-1)), Magnesium sulfate hydrate (MgSO$_4$, 22189-08-8), HEPES (7365-45-9), Dulbecco's modification of Eagle's medium

(EMEM), fetal bovine serum (FBS), serum-free medium (SFM), hydrogen chloride (HCl, 7647-01-0), ethanol (64-17-5, 190 proof), fluorescent LAMP kit (WarmStart, New England BioLabs, NEB), RNA purification kit (Monarch, NEB), bovine serum albumin (BSA), microfluidic chips (QuantStudio 3 Digital PCR 20k Chip Kit V2, Applied Biosystems), nuclease-free water and Amicon™ ultra centrifugal filter units (UFC510024) were purchased from Fisher Scientific. Aminated $SiO_2$ beads in ethanol (SIAN500-25M) were purchased from Nanocomposix. Palivizumab/Synagis was a gift from University of Texas Southwestern Medical Center or obtained from MedImmune, Gaithersburg, MD, CAT# NDC60574. HEp-2 cells (CAT# CCL-23) and RSV RNA (VR-1540DQ) were purchased from American Type Culture Collection (ATCC). Viral RNA extraction buffer and LAMP primers were ordered from Sigma-Aldrich. All aqueous solutions were prepared using deionized water ($dH_2O$) with a resistivity of 18.0 MΩ·cm.

## Methods

*Nanoparticle synthesis and characterization.* Gold nanoparticle (AuNP) seeds were first synthesized using classical Plech Turkevich method with slight modifications[43]. Briefly, 1 mL of $HAuCl_4$ aqueous solution (25 mM) was added to a clean 250 mL Erlenmeyer flask with 98 mL of pure water and allowed to boil on a hot plate with magnetic stirring. Subsequently, 1 mL of $Na_3CA·2H_2O$ (112.2 mM) aqueous solution was quickly injected into the boiling solution with a pipette. The solution was kept stirring and boiling for 10 min until its color turned red. After cooling the solution to room temperature, $dH_2O$ was added to bring the volume to 100 mL. The products were stored in the dark at room temperature for future use.

AuNPs with larger sizes were synthesized according to a seed-mediated growth method with slight modifications, where the above-mentioned AuNPs were employed as the seeds. In brief, the appropriate amount of pure water, $HAuCl_4$ precursor, $Na_3CA·2H_2O$, and AuNP seeds were added orderly to a clean 250 mL flask under vigorous magnetic stirring at room temperature according to Table S5. Then the reducing aqueous solution containing a certain amount of hydroquinone was injected rapidly into the flask. The solution immediately switched color to purple and then to red in a few minutes. The reaction was left overnight at room temperature. Final products were stored in the dark at room temperature for future use. Note that all the particles' concentrations in this study were determined based on the size-dependent empirical formula with a combination of UV-vis measurement and transmission electron microscope (TEM)[28].

*Biosafety statements.* The research project was approved and performed strictly in adherence to CDC/NIH guidelines and the experimental protocols were approved by the University of Texas at Dallas Institutional Biosafety & Chemical Safety Committee and the University of Texas Southwestern Medical Center Biosafety Committee. Human Metapneumovirus (hMPV, CAT# 0810163CF) and Parainfluenza virus type 1 (PIV, CAT# 0810014CF) were purchased from ZeptoMetrix, and H1N1 influenza virus type A (IVA, CAT# IHA-003) was purchased from ProSpec-Tany TechnoGene Ltd. Human respiratory syncytial virus (RSV) A2 strain (CAT # VR-1540) was purchased from ATCC.

*Conjugation of AuNPs and Synagis as detection probes for RSV.* Conjugation of AuNPs and Synagis was adapted from a previous report[44]. In brief, 5 mM DTSSP as a cross-linker was reacted with primary amines of Synagis at a molar ratio of 125:1 in 2 mM membrane borate buffer first, then followed by an overnight membrane dialysis process to eliminate free DTSSP before concentrated using 100 kDa Amicon centrifugal filter. The resulted product (DTSSP-Syn) was then mixed with 15 nm AuNPs suspension in 2 mM borate buffer at a molar ratio of 500:1 and incubated overnight at 4 °C. Afterward, the products (15 nm AuNP probes) were washed three times with 2 mM borate buffer and re-dispersed in 2 mM borate buffer, and then kept at 4 °C for storage before further testing.

As for BSA backfilling, we utilized DTSSP as a covalent linker to modify BSA with a molar ratio of 125:1 in 2 mM borate buffer, and then incubated the 0.1% of DTSSP-BSA with 15 nm AuNPs probes on the ice bath for 1 h. The products (BSA-backfilled AuNPs probes) were washed three times using 2 mM borate buffer and re-dispersed in 2 mM borate buffer. The samples were kept at 4 °C for storage before further use.

*Large scale propagation of RSV A2 in HEp-2 cells.* RSV A2 strain was propagated in HEp-2 cells. HEp-2 cells were cultivated in EMEM/5% FBS and infected at a multiplicity of infection (MOI) of 0.01 to minimize the production of defective interfering particles. Viral working stocks were prepared using 30–60% (w/v) non-continuous sucrose density centrifugation (Beckman Coulter SW-28 rotor, 28,000 rpm, 4 °C for 90 min). The virus-containing band at the 30–60% interface was collected (Fig. S23), distributed into aliquots, and stored at −80 °C[45]. Viral titers were determined by endpoint dilution assay as previously described[46]. Briefly, serial dilutions of RSV stock were prepared and inoculated onto Hep-2 cells in 96-well plates (Fig. S24). After 6 days of incubation at 37 °C and 5% $CO_2$, plates that displayed cytopathic effects were visualized by staining with crystal violet dye. Cells that undergo infection and death lose the adherence and were subsequently washed away, reducing the amount of crystal violet staining in the culture plate. The plaque-forming unit of viral working stocks was calculated based on the median tissue culture infectious dose (TCID50) to quantify the RSV infectivity titer.

*Detection of viruses using the AuNP-based probes.* Purified RSV and other closely related respiratory viruses such as hMPV, PIV, and IVA (used as received) were incubated with AuNPs-Synagis probes for at least 30 min at room temperature in the 2 mM borate buffer. The completed assay solutions were subjected to colorimetric measurements or DIAMOND tests.

*Detection of virus in complex sample matrix using the BSA-backfilled AuNPs probes.* We used BD Universal Viral Transport Collection Kits to collect nasal swab samples from the nostrils of healthy adults (a group of 3 males, age 30). The soft end of the swab was inserted into each nostril and slowly rotated for at least 30 s to ensure as much nasal discharge as possible per adult. Then the swab was placed into a sterile tube filled with viral transport medium (VTM) and sealed tightly. Mechanical homogenization of collected samples was processed upon 5 min of high-speed vortexing with small glass beads in the sterile tube. The unpurified virus suspensions including RSV, hMPV, PIV, and IVA were then spiked into the as-prepared nasal swab healthy control samples and used as clinical samples. VTM was used as an additional negative control. The BSA-backfilled AuNPs probes were directly incubated with those virus-spiked samples with a volume ratio of 2:1 for 30 min prior to the measurements.

The use of human nasal swab samples was approved by Institutional Review Board (IRB) at University of Texas at Dallas (ID: 20MR0093).

*Plasmonic nanobubble generation and detection.* The plasmonic nanobubble (PNB) generation was conducted using ultrafast pump laser pulses (28 ps, 532 nm, PL2230, Ekspla). The AuNPs suspension was pushed through a 200 μm microcapillary (VitroTube, 8320) using a syringe pump (New Era Pump Systems Inc.) and irradiated by a pump laser. The laser fluence was adjusted by rotating the beam attenuator (ThorLABs, VBA05-532) and was measured using a laser power meter (FieldMaxII-TOP, Coherent). PNB signals were monitored by a continuous laser as probe beam (Red HeNe laser, 633 nm, R-30989, Newport) and its intensity was recorded with a photodetector (FPD510-FV, Thorlabs) and an oscilloscope (LeCroy WaveRunner204Xi-A).

For the 75 nm AuNPs detection, serial suspensions with concentrations equaling λ = 80, 40, 27, 8, 4, 2.7, 0.8, 0.4, 0.27, 0.08, 0.04 were prepared and tested at a laser fluence of 10,000 mJ/cm$^2$. All the measurements were recorded one minute per sample (3000 pulses at 50 Hz).

For the size differentiation assay, serial suspensions of AuNPs with 15, 35, 50, and 75 nm in diameter and $1.5 \times 10^{10}$ mL$^{-1}$ (λ = 240) in concentration were prepared and tested under a laser fluence of 2000 mJ/cm$^2$.

For the mixture sample test, the particle concentrations of two AuNP suspensions were set the same as $1.5 \times 10^{10}$ mL$^{-1}$ (λ = 240) and $3.125 \times 10^7$ mL$^{-1}$ (λ = 0.5) for 15 and 75 nm AuNPs, respectively. The PNB detection was performed at a laser fluence of 7500 mJ/cm$^2$.

For the $SiO_2$ beads detection, the as-purchased $SiO_2$ beads in ethanol were first centrifuged and washed with $dH_2O$ twice and re-dispersed in 2 mM citrate-HCl buffer (pH = 3) with a particle concentration of $1 \times 10^{10}$ mL$^{-1}$. The $SiO_2$ beads were sonicated and vortexed before mixing with 15 nm citrate-AuNPs suspension. The mixture was incubated for 30 min at room temperature before further tests. The PNB detection was performed at a laser fluence of 3000 mJ/cm$^2$.

For the virus detection, the completed assay solutions were directed to the DIAMOND test at a laser fluence of 3000 mJ/cm$^2$.

*Digital LAMP (dLAMP) reaction.* For the synthetic RSV RNA detection, we mixed 10 μL of LAMP master mix, 2 μL of 10x primer mix (Table S3), 0.5 μL of fluorescent dye, and 7 μL of RNA template, and 1.5 μL of nuclease-free water at room temperature. We then loaded 15 μL of the mixture into the microwell array chip and subsequently sealed it with oil. The chip was incubated at 65 °C in a heating block for 30 min and subjected to fluorescent imaging using a slide scanner (Olympus VS 120) with the ×2 objective.

For the RSV detection, 10 μL of RSV sample was mixed with 5 μL of extraction buffer at room temperature for 10 min and subjected to the purification following the manufacture's instruction. The final product was dispersed in 40 μL elution buffer and stored in −20 °C before use. For the RNA detection, we mixed 10 μL of LAMP master mix, 2 μL of 10x primer mix, 0.5 μL of fluorescent dye, and 7 μL of RNA extracts, and 1.5 μL of nuclease-free water at room temperature. We then loaded 15 μL of the mixture into the microwell array chip and subsequently sealed it with oil. The chip was incubated at 65 °C in a heating block for 30 min and subjected to fluorescent imaging using a slide scanner with the ×2 objective.

*MATLAB for PNB detection.* The raw data of PNB signals were processed by a customized MATLAB script[47]. The major functions of the script include pre-filtration, PNB signal recognition by signal-to-noisy-ratio (SNR), parameters extraction, threshold calculation, and frequency counting for "on" signals.

*MATLAB for fluorescent image analysis.* The fluorescent images of dLAMP detection were processed by a customized MATLAB script. The major functions of the script include image reading, determining the signal of microwells via pattern recognition, threshold calculation for each well, and frequency counting for "on" signals.

*Characterization techniques and data analyses.* The absorbance of samples in microtiter plates was read using microplate reader (Synergy 2, BioTek). The dynamic light scattering (DLS) was measured using Malvern ZetaSizer Nano ZS. Extinction spectra were obtained with a spectrophotometer (DU800, Beckman Coulter). The TEM images were taken using a JEOL JEM-2010 microscope operated at 120 kV. The pH values of buffer solutions were measured using a pH Meter (Accumet AP71). All data were collected in no less than triplicate and reported as mean and standard deviation for statistical analysis. A two-sample *t*-test assuming equal variance in Origin software was conducted to determine *p* values and statistical significance.

*Poisson statistics.* The theoretical prediction by Poisson statistics in our study was calculated based on Eq. (2). Specifically, the data in Fig. S4 was generated using Microsoft Excel Worksheet (Data-Data Analysis-Random Number Generation-Poisson).

**Reporting summary**. Further information on research design is available in the Nature Research Reporting Summary linked to this article.

## Data availability

Datasets supporting the findings of this work are available within the paper and its supplementary information files. A reporting summary file for this article is available. Additional information for research purposes is also available from the corresponding authors upon request. Source data are provided with this paper.

## Code availability

The software used in this study is described in the "Methods" section. MATLAB codes for PNB signal analysis and dLAMP image analysis are available at https://doi.org/10.5281/zenodo.5708858. Additional information is available from the corresponding authors upon request.

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

## Acknowledgements

This research is partially supported by the National Institutes of Health (NIH) grants R21AI140462 and R01AI151374 and the CDMRP Peer Reviewed Medical Research Program Discovery Award PR192581 to Z.Q. The content is solely the responsibility of the authors and does not necessarily represent the official views of the funding agencies. We are grateful to the undergrad UT design team (Marisa Romero, Adrienne Phillips, Vinit Sheth, John Yoo, Michelle Chang, Austin Daneman) for helping with benchtop device assembly. We would like to thank Dr. Abdullah Bayram for useful discussions in further optimization of the optofluidic system to improve the sampling efficiency for the DIAMOND technique. We also thank Dr. Ruth Levitz and Xiaoqing Li for providing insightful suggestions for RSV quantification.

## Author contributions

Y.L. and H.Y. carried out the experiment and performed the data analysis and contributed equally to this work. Y.L., H.Y., and H.H. prepared the virus samples. Y.L., P.K., and C.X. developed the MATLAB script. Y.L., H.Y., and Z.Q. wrote the manuscript. Z.Q. and J.S.K. conceived the original idea and supervised the project. H.Y. helped supervise the project. All authors revised the manuscript and have given approval to the final version of the manuscript.

## Competing interests

Y.L., Z.Q., H.Y., and J.S.K. are the inventors on a provisional patent related to this work filed by University of Texas at Dallas. Z.Q. and J.S.K. hold equity interest in Avsana Labs, Incorporated, which aims to commercialize the DIAMOND technology. The remaining authors declare no competing interests.
