## [Peer Review File · Nature Communications]

Digital Plasmonic Nanobubble Detection for Rapid and Ultrasensitive Virus DiagnosticsEditorial Note: Parts of this Peer Review File have been redacted as indicated to maintain the confidentiality of unpublished data.

REVIEWER COMMENTS

Reviewer #1 (Remarks to the Author):

This manuscript describes the detection of gold nanoparticles by digital counting of laser-induced microbubbles. The approach is unlikely to be practical because of the need for a laser and complex instrumentation. The workflow reported in the manuscript is simple because it only detects purified particles (SiO₂ or purified RSV) but these are contrived systems that don't represent real-world samples, which will have non-specific binding (NSB) issues. The authors' statement in the Introduction "digital assays have suffered from complex assay operations" is precisely because sample preparation is required to remove NSB. For any real assay, it will require the same "complex assay operations" of multiple washings, which will require pelleting, filtering, or other separation of some type. The comparison with a lateral flow assay, which is not particularly sensitive, is fine but a complex sample will undoubtedly give results that are not as good. There are other microbubble methods that use much simpler implementation: recent preprint can be found here <https://www.medrxiv.org/content/10.1101/2021.03.17.21253847v1> (many other examples). On a more positive note, the work performed is well done with proper validation but the use of contrived purified samples significantly diminishes the impact.

Overall, a new method to detect nanoparticles is reported but the practicality and potential for diagnostics is limited both because of the complex instrumentation required and the lack of testing with realistic samples.

Reviewer #2 (Remarks to the Author):

Q. What are the noteworthy results?

The authors demonstrate a novel strategy for the rapid and sensitive detection of the respiratory syncytial virus. The strategy is based on plasmonic nanobubbles, generated by laser heating of gold nanoparticles and detected through an optofluidic setup. The authors demonstrate (1) the possibility to detect single plasmonic nanoparticles and discriminate nanoparticles of different sizes, (2) to identify heterogeneities in nanoparticle suspensions, (3) the sensitive and absolute quantification of a model-system for a homogeneous assay and (4) the sensitive detection of intact respiratory syncytial virus.

Q. Will the work be of significance to the field and related fields? How does it compare to the established literature? If the work is not original, please provide relevant references.

The use of plasmonic nanobubbles for single-molecule detection is, to the best of my knowledge, an overall novel strategy which opens up interesting possibilities for future developments in single-molecule detection systems. However, I believe some additional information or positioning is required to enable proper assessment of the presented technique and proper comparison with existing techniques.

- In the introduction, the authors first describe the emergence of digital assays and refer to the technique of Rissin et al. Afterwards, the authors point towards further innovations based on micro/nano-particles. It is not clear to me why they make that distinction, since the technique of Rissin et al. also involves microparticles. Also, from then onwards, the term single-particle digital assay is introduced. It is not clear to me if what the authors refer to with this 'single-particle' term. If they refer to the single target particle to be analyzed, that is already included in the definition of digital bioassay and hence redundant. If they refer to the micro/nanoparticles used in the assay, their technique does not fit this term as they rely on multiple nanoparticles to label a single RSV for detection. A clarification or change of terminology is required. On the same note, the title refers to a single plasmonic nanobubble. It is not clear to me if this implies that multiple vapor bubbles of a number of 15 nm nanoparticles, clustered on an RSV, merge into a single big vapor bubble? An explanation or discussion on this topic would be good for clarity.

- Whereas the authors indicate a number of novel techniques for single-molecule detection, I feel there is a lack of positioning of DIAMOND with respect to flow cytometry-based systems, including 'flow virometry'. Both strategies are very much alike: both rely on detection of a target in a stream passing a laser beam, where in flow cytometry the viral particles are labelled with fluorophores for detection rather than nanoparticles, as is the case for DIAMOND. Because of their resemblance, the comparison of DIAMOND with flow cytometry in the introduction and in the discussion (in terms of sensitivity, specificity and detection time) is missing and should be provided to enable a proper assessment of the relevance of this technique.

- The authors position their technique as a technique for single-molecule detection and compare the results to a colorimetric and LFA assay. However, the DIAMOND system is more complex and does not allow self-testing by the patient. Therefore, it would be interesting to position their sensitivity with respect to existing techniques of similar complexity or higher complexity, including digital ELISA and PCR. It would be interesting to see how this novel approach performs when compared to those established techniques.

- The technique, demonstrated by the authors is highly interesting for the detection of single viruses, vesicles and cells. However, I fail to see the applicability for the detection of smaller entities such as proteins or DNA molecules. The approach requires clustering of nanoparticles and as such, if a small protein is to be detected, it will most likely be labelled with only one or very few nanoparticles. According to my understanding, the proposed technique will face its limits for the detection of such molecules and the alternative techniques described in the introduction might be of more use there? A discussion on this matter would be really insightful.

Q. Does the work support the conclusions and claims, or is additional evidence needed?

The authors carried out a wide variety of experiments to characterize the system, substantiated by the extensive body of supplementary information. However, I would like to raise a few questions that should be tackled either in additional discussions or via additional evidence.

- The detection volume is rather big (16 pL), substantiated by the highly relevant comment of the authors about the sampling efficiency. Why did the authors not match the dimensions of the capillary and laser-beam? Are such dimensions not compatible with capillaries? A short note in this respect would be helpful in clarifying this.
- The authors demonstrate that DIAMOND allows size differentiation by showing the efficient discrimination of nanoparticles of 15 nm up to 75 nm particles. It is not clear why the authors chose the 15 nm particles for detection of the SuO₂ beads and the RSV.
- The authors demonstrate that DIAMOND allows identification of heterogeneities in a nanoparticle suspension. However, it is not clear to me why this is relevant in the context of the detection of viral particles. If an RSV particle is labelled with nanoparticles, a cluster of 15 nm nanoparticles arises. This cluster of 15 nm particles should be detected in a background of free-floating 15 nm nanoparticles, rather than a 75 nm nanoparticle in a background of 15 nm particles. Hence the detection of bigger nanoparticles might not be as relevant for the final application as the detection of clusters. It would be interesting to explain the rationale behind this in more detail.
- The authors refer to their technique as a system that enables fast measurements (within 2 minutes). However, the incubation of RSV with AuNP probes of 30 min does not seem to be included in that time-frame. This is to be clarified as it might be perceived as misleading when comparing DIAMOND to other techniques.
- It would be highly interesting to include a short discussion on the multiplexing possibilities of this technique, as this is of great importance to the field and thus highly relevant.
- The linear dynamic range depicted in Fig 4e is small and as such, samples in only a very narrow concentration range can be detected reliably. Is this a given limitation of the system or is there a workaround for the future? If not, how do the authors foresee to deal with that for testing of real samples?
- Figure 3b: Please indicate i, ii and iii. Also, the caption described a red arrow but the arrow is grey.

Q. Are there any flaws in the data analysis, interpretation and conclusions? - Do these prohibit publication or require revision?

- Figure 4b: Please indicate where the RSV is. If I understand it correctly, I see clusters of nanoparticles aside of the RSV (on Fig 4b(i) at right bottom side, in 4b(ii) at top center?). How will these clusters aside of the RSV reflect in a signal in the DIAMOND? Will these clusters remain attached to the RSV or will they result in free-floating clusters and thus additional signals? Overall, it is not clear to me how the authors prevent or anticipate aggregation of NPs in the sample and a brief discussion on this topic is needed.

Q. Is the methodology sound? Does the work meet the expected standards in your field?

- The authors demonstrate the use of DIAMOND in transporting medium. However, the performance of the system in nasal swabs of healthy controls, spiked with RSV is required to enable proper assessment of the proposed technique and to meet the standards in the field.

Q. Is there enough detail provided in the methods for the work to be reproduced?

- The methods are clear and include sufficient detail to reproduce the work.

Reviewer #3 (Remarks to the Author):

The authors report on a particle-based biosensing assay that uses bubbles generated by photothermal heating of nanoparticle assemblies. The presence of analyte leads to assembly of plasmonic particles on a scaffold, e.g. a virus particle. The plasmonic assemblies are flown through a fluidic channel and illuminated by a pulsed laser, causing nanobubble generation. This provides a non-linear readout that is strongly biased to the assemblies allowing for the counting of the number of assemblies against a background of free particles in solution. The authors show several benchmarking experiments meant to optimize the particle size and concentration, and finally show the application to a single-step immunoassay for virus detection. I think the assay format is well presented and tested. I believe the manuscript would profit from additional measurements and a fairer comparison to the state-of-the-art to allow for a better description of the assay performance. In particular:

- although the assay is new as far as I know, the manuscript contains a rather lengthy methodological description with little actual sensing data. It would have been nice to show the ability to detect multiple analytes, but I can imagine this is not an easy feat at this stage of the technology. Importantly though, controls are shown in figure 5 for colorimetric detection, but the controls in the nanobubble assay are not mentioned. The controls should be performed and shown in Figure 5e for non-complementary gold particles or different virus types to demonstrate the specificity of the detected signals.

- the sensitivity of the assay is now compared to lateral flow and colorimetric assays, which are indeed simple but do not provide state-of-the-art sensitivity. A fairer comparison in terms of sensitivity would be to involve commercially available digital assays (e.g. ELISA, PCR, and others, see e.g. Lab Chip 2020, 20, 2816-2840). Particularly the comparison to flow cytometry approaches is relevant as the setup design is similar. These commercially available approaches are now mentioned

in passing in the introduction and in the supplementary material, but a quantitative comparison is needed to judge the benefits of the presented method.

- the authors claim in the introduction that other digital assays are "complex". It is not specified what the complexity is they refer to, and in what sense the current assay is easier.

- The discussion now heavily focuses on the pulse frequency to increase the speed of the assay. It would be nice to increase the scope of the discussion. For example: the presented assay requires a pulsed and a CW laser that are overlapped in their focus. This is not a cheap setup and not easily achieved in other labs. Can the authors discuss future improvements needed to bring the assay to a broad range of labs? Also, is the method easily extended to other analytes, and if so, which ones are most obvious candidates?

minor comments:

- on page 4 on line 23 it is stated that the theoretical probability is predicted by Poisson statistics but it is not explained how.

- on page 5 at the top it is stated that the distribution profiles could be fitted with 1.98 and 3.51-order dependencies. Why is this indicative of a mechanism involving nanobubbles?

- in figures 2b/2c/4e/4f/5c/5e: what are the red lines?

- in Fig 5b it is unclear to me what we observe here, I see fibrous material with gold particles on top, is that expected?

- the main text discusses little about the fluidics, e.g. the design of the microfluidic channels, the flow speed, spot size etc. It would be nice to move certain aspects from the SI to the main text to better describe the method.

Response Letter to Reviewers

Reviewer #1 (Remarks to the Author):

This manuscript describes the detection of gold nanoparticles by digital counting of laser-induced microbubbles. The approach is unlikely to be practical because of the need for a laser and complex instrumentation.

Response: We thank the reviewer for bringing up this concern. We agree that our current setup for DIAMOND test is bulky using a laboratory-level laser system. As an ongoing effort, we have tested a portable nanosecond laser (Wedge-HB-532, RPMC) as the pump laser. It has dimension of $26 \times 22 \times 8 \text{ cm}^3$ and can be integrated in a benchtop device ($15 \times 15 \times 6 \text{ inch}^3$, **Fig. S17** of revised Supporting Information). In the nanoparticle detection, this type of laser generates stable plasmonic nanobubble signals at high particle concentration and discrete signals similar to that of bulky laser (**Fig. S18** of revised Supporting Information). Furthermore, there have been reports in small portable devices based on thermal lens, a different mechanism but with similar complexity (Lab Chip, 2011, 11, 2990-2993). We believe the miniaturized device is feasible and would find broad applications for diagnosis. We have added a discussion on this topic and highlighted it in Discussion of revised manuscript (Page 9).

Information about Wedge-HB-532, RPMC available at:
<https://www.rpmclasers.com/product/wedge-b-532/>

Changes on Page 9:

“First, it is possible to design a small benchtop device for the plasmonic nanobubble (PNB) measurements that can be distributed to other labs (38). By replacing the research-grade picosecond pulse laser with a smaller nanosecond (ns) laser (e.g., Wedge-HB-532, RPMC), all components can be integrated in a benchtop device ($15 \times 15 \times 6 \text{ inches}$, **Fig. S17**). Evaluation of PNB generation and detection by this ns laser shows robust results across a range of AuNP concentrations (**Fig. S18**). More importantly, the ns laser provides a repetition rate up to 2,000 Hz and allows a much faster readout and thus more efficient event counting.”

Fig. S17. Design of benchtop device integrating DIAMOND. The device has a dimension of 15×15×6 inches.

Fig. S18. Detection of 75 nm AuNPs by DIAMOND using a nanosecond laser (Wedge-HB-532, RPMC). (a) Representative PNB signal traces (100 pulses) for 75 nm AuNP suspensions with different particle concentrations. (b) Bivariate plots of amplitude (Ampl) and AUC extracted from 3,000 pulses for the three samples in (a). (c) Bar plot of experimental frequencies (f_{on}) as determined in (b) and theoretical probability predicted by Poisson statistics for the given λ .

The workflow reported in the manuscript is simple because it only detects purified particles (SiO₂ or purified RSV), but these are contrived systems that don't represent real-world samples, which will have non-specific binding (NSB) issues. The authors' statement in the Introduction "digital assays have suffered from complex assay operations" is precisely because sample preparation is required to remove NSB. For any real assay, it will require the same "complex assay operations" of multiple washings, which will require pelleting, filtering, or other separation of some type.

Response: We thank the reviewer for pointing out this issue. To verify the ability of DIAMOND in real-world sample test, we spiked the virus in nasal swab samples that were collected from healthy controls and tested it. Utilizing bovine serum albumin (BSA) backfilled AuNP-based probes, DIAMOND can specifically detect unpurified RSV among different respiratory viruses with a detection limit of 102 PFU/mL, similar to the purified sample test. Results were provided in the new **Fig. 6** of revised manuscript. On the other hand, the sample washing is not required in the homogeneous immunoassays because the target-specific signal can be differentiated from the free labels. Other examples include the Brownian-motion based digital homogeneous immunoassays (ACS Nano 2019, 13, 13116-13126 and Nat. Commun. 2018, 9, 2541). Therefore, one of the merits for DIAMOND is its implementation on homogeneous immunoassay that bypasses the need for sample washing and simplifies the assay implementation. We have emphasized this content in the revised manuscript.

Fig. 6. Detection of respiratory viruses spiked in nasal swab samples by DIAMOND. (a, c, e) Bivariate plots of amplitude and AUC extracted from 3,000 pulses for the assay solutions incubating different respiratory viruses with (a) AuNP-Synagis as probes, (c) bovine serum albumin (BSA)-backfilled AuNP-Synagis probes, and (e) RSV of different titers (PFU/mL) with BSA-backfilled AuNP-Synagis probes. (b, d, f) Corresponding f_{on} counted from (a, c, e) against (b, d) different respiratory viruses and (f) RSV titers, respectively. VTM is viral transportation medium. Insets in (a, c, e) show the model of AuNP-based probes used correspondingly. Inset in (f) shows the linear detection range and as-determined LOD .

The comparison with a lateral flow assay, which is not particularly sensitive, is fine but a complex sample will undoubtedly give results that are not as good. There are other microbubble methods that use much simpler implementation: recent preprint can be found here <https://www.medrxiv.org/content/10.1101/2021.03.17.21253847v1> (many other examples).

Response: We thank the reviewer for pointing out this issue. We refer to the previous comment on testing complex samples and demonstrated the capability to test virus spiked in nasal swab

samples. Also, we performed further experiments to compare with single-molecule digital assays as discussed below. The microbubble method is an interesting advance beyond the current digital assays.

In the revised manuscript, we compared DIAMOND with digital loop-mediate isothermal amplification (dLAMP). dLAMP is a rapid molecule test and provides absolute quantification of nucleic acids, which has been used to quantify a variety of viruses. We used commercially available LAMP kit and microwell chip for the reaction and designed a MATLAB code for data analysis. Below figures are the dLAMP detection results of synthetic RNA (**Fig. S15** in revised Supporting Information) and RSV extracts (**Fig. S16** in revised Supporting Information). Based on the results, we can see the dLAMP has detection limit of ~200 PFU/mL RSV, similar to that of DIAMOND. Also, we estimated that 100 PFU/mL is equivalent to 1 copies/ μ L RNA, based on the dLAMP results. That is to say, DIAMOND has a LOD of single-molecule detection, which is competitive with nucleic acid amplification methods. In addition to the comparable sensitivity with digital assays, DIAMOND can detect intact virus without nucleic amplification and additional liquid handling associated with the assay (i.e., virus extraction, thermal incubation, and chip loading). Therefore, we believe DIAMOND has a simplified diagnostic approach and represents an important advance for virus detection. We have added this content in the revised manuscript (Page 8).

Changes on Page 8:

“To compare DIAMOND with other state-of-the-art methods, we performed measurements using digital loop-mediated isothermal amplification (dLAMP). dLAMP is a rapid molecule test and provides absolute quantification of nucleic acids, which has been used to quantify a variety of viruses (33-36). To perform the dLAMP, we used commercially available fluorescent LAMP kit and microwell chips for detecting genomic RNA of RSV (A2 strain) as target. A set of primers (**Table S3**) were used according to a previous publication (37). **Fig. S15a-e** show the fluorescence images of the dLAMP results after incubating at 65 °C for 30 min, where the number of positive wells (brighter) decreases with the concentration of target RNA. A calibration curve ($R^2=0.9998$, **Fig. S15f**) was established as a function of the RNA inputs. The *LOD* was estimated to be 2 copies/ μ L using synthetic RNA (inset of **Fig. S15f**). We then used this method to detect RSV from the spiked samples. The RSV extracts were collected using commercially available RNA extraction buffer and purification kit. **Fig. S16a-d** show the dLAMP detection results for RSV extracts at different concentrations. Specifically, RSV of ~200 PFU/mL can be detected (**Fig. S16e**), similar to that of DIAMOND. Furthermore, referring to the calibration curve in **Fig. S15f**, we estimated that 100 PFU/mL is equivalent to 1 copies/ μ L RNA with our protocols (**Fig. S16f**). In other words, DIAMOND has a *LOD* of single-molecule detection, which is competitive with nucleic acid amplification methods.”

Fig. S15. Detection of RSV RNA via digital loop-mediated isothermal amplification (dLAMP). (a-e) Fluorescence images of the microwell chips after dLAMP with varied RNA inputs. All images have intensity range of 0-10,000 relative fluorescence units (RFU). (f) A plot of the frequency of positive wells (f_{on}) against different RNA inputs (copies per microliter). The positive wells were determined as the maximum fluorescence intensities larger than a threshold of three standard deviations above the mean. Inset shows the limit of detection (LOD), where the background is set as 3-times standard deviation above the zero calibrator. Each well of the microfluidic chip has a volume of 750 pL.

Fig. S16. Detection of RNA extracts from RSV spiked samples via dLAMP. (a-d) Fluorescence images of the microwell chips after dLAMP with varied RSV inputs. All images have intensity range of 0-10,000 RFU. (e) A plot of f_{on} against RSV titers. The background is set as 3-times standard deviation above the zero calibrator. (f) Concentration of RNA detected in RSV extracts based on the background-subtracted frequency (f_{on}') referring to the calibration curve from **Fig. S15f**.

On a more positive note, the work performed is well done with proper validation but the use of contrived purified samples significantly diminishes the impact. Overall, a new method to detect nanoparticles is reported but the practicality and potential for diagnostics is limited both because of the complex instrumentation required and the lack of testing with realistic samples.

Response: We thank the reviewer for the positive comment and summary of our pros and cons. As discussed above, we performed additional experiments to demonstrate the ability to detect complex samples (e.g., unpurified virus spiked in nasal swab sample).

In terms of the potential of using a benchtop device, we believe the proposed DIAMOND technique would have practical applications and serve as a complementary to the current digital immunoassays for virus detection.

We appreciated all the comments provided from the reviewer that help us improve the quality of our manuscript.

Reviewer #2 (Remarks to the Author):

Q. What are the noteworthy results?

The authors demonstrate a novel strategy for the rapid and sensitive detection of the respiratory syncytial virus. The strategy is based on plasmonic nanobubbles, generated by laser heating of gold nanoparticles and detected through an optofluidic setup. The authors demonstrate (1) the possibility to detect single plasmonic nanoparticles and discriminate nanoparticles of different sizes, (2) to identify heterogeneities in nanoparticle suspensions, (3) the sensitive and absolute quantification of a model-system for a homogeneous assay and (4) the sensitive detection of intact respiratory syncytial virus.

Response: We thank the reviewer for the great summary of our work.

Q. Will the work be of significance to the field and related fields? How does it compare to the established literature? If the work is not original, please provide relevant references.

The use of plasmonic nanobubbles for single-molecule detection is, to the best of my knowledge, an overall novel strategy which opens up interesting possibilities for future developments in single-molecule detection systems. However, I believe some additional information or positioning is required to enable proper assessment of the presented technique and proper comparison with existing techniques.

Response: We thank the reviewer for the encouraging comments and have addressed each comment below.

- In the introduction, the authors first describe the emergence of digital assays and refer to the technique of Rissin et al. Afterwards, the authors point towards further innovations based on micro/nano-particles. It is not clear to me why they make that distinction, since the technique of also involves microparticles.

Response: We thank the reviewer for pointing out this issue. We made such distinction because the functionalities of micro/nanoparticles have been changed along with the development of digital immunoassays. In the conventional dELISA by Rissin et al., the microparticles serve as carriers that facilitate the sample washing and partition. While in recent innovations, those micro/nanoparticles also serve as signal reporters that can generate detectable signals without the need for signal amplification.

Also, from then onwards, the term single-particle digital assay is introduced. It is not clear to me if what the authors refer to with this 'single-particle' term. If they refer to the single target particle to be analyzed, that is already included in the definition of digital bioassay and hence redundant. If they refer to the micro/nanoparticles used in the assay, their technique does not fit this term as they rely on multiple nanoparticles to label a single RSV for detection. A clarification or change of terminology is required.

Response: We are referring to single-particle, or single-nanoparticle as our detection method, since our assay detects the single nanoparticles (e.g., **Fig. 2** and **4**) and single viral particles (e.g., **Fig. 5**). The single particle counting and analysis (amplitude, AUC) also serve as the basis of our technology. In this regard, the single-particle analysis indeed is single-analyte analysis, same as the reviewer's comment. Therefore, we only keep the term "single-nanoparticle detection" in Page 3 and 4 of revised manuscript, because DIAMOND does allow single AuNP detection and may have additional implications as an analytical tool for NPs.

Changes on Page 3:

"Taking advantage of these unique properties, we designed an optofluidic setup to flow the NP suspension in a micro-capillary for single-NP detection (Fig. 1a** and **Fig. S1**)."**

Change on Page 4:

"DIAMOND allows single-nanoparticle (NP) detection and size differentiation"

"We first evaluated the ability of DIAMOND for single-NP detection..."

On the same note, the title refers to a single plasmonic nanobubble. It is not clear to me if this implies that multiple vapor bubbles of a number of 15 nm nanoparticles, clustered on an RSV, merge into a single big vapor bubble? An explanation or discussion on this topic would be good for clarity.

Response: The reviewer raised a good point. The plasmonic interactions between the AuNP upon binding to the virus suggest that the distance between them is short. There is no direct evidence to

suggest that the nanobubbles are merged together or stay as multiple vapor bubbles, however we can make some estimations based on the data we have. The bubbles for clustered AuNPs have a larger amplitude and longer duration (**Fig. S10**). This result suggests that there may be some combination of small bubbles, as simply adding multiple small bubbles would only increase the amplitude but not the bubble duration. To avoid confusion, we remove the description of “single-nanobubble” detection in the title.

- Whereas the authors indicate a number of novel techniques for single-molecule detection, I feel there is a lack of positioning of DIAMOND with respect to flow cytometry-based systems, including ‘flow viometry’. Both strategies are very much alike: both rely on detection of a target in a stream passing a laser beam, where in flow cytometry the viral particles are labelled with fluorophores for detection rather than nanoparticles, as is the case for DIAMOND. Because of their resemblance, the comparison of DIAMOND with flow cytometry in the introduction and in the discussion (in terms of sensitivity, specificity and detection time) is missing and should be provided to enable a proper assessment of the relevance of this technique.

Response: The reviewer raised an interesting point. As suggested, we have included a comparison between those two methods in the revised manuscript (Page 3-4 and 9).

Changes on Page 3-4:

“Since the PNBs are transient events and has no cross-talk between laser pulses, it allows counting “on” and “off” signal in a compartment-free manner (**Fig. 1b**). Although its detection scheme is similar to the flow cytometry, DIAMOND utilizes non-fluorescent labels for the signal generation and detects analytes without flow focusing.”

Changes on Page 9:

“Taking advantage of the high-throughput ns laser and focusing flow, we expect to increase the event counting (e.g., 1 million readings within 10 minutes, similar to flow cytometer).

- The authors position their technique as a technique for single-molecule detection and compare the results to a colorimetric and LFA assay. However, the DIAMOND system is more complex and does not allow self-testing by the patient. Therefore, it would be interesting to position their sensitivity with respect to existing techniques of similar complexity or higher complexity, including digital ELISA and PCR. It would be interesting to see how this novel approach performs when compared to those established techniques.

Response: We agree with the reviewer that a comparison between DIAMOND and other digital assay is needed to clarify the advantages. Therefore, we have conducted the digital loop-mediate isothermal amplification (dLAMP) for comparison. dLAMP is a rapid molecule test and detects the amplified DNA via fluorescent microscope. We used commercially available LAMP kit and

microwell chip for the reaction and designed a MATLAB code for data analysis. Below figures are the dLAMP detection results of synthetic RNA (**Fig. S15** in revised Supporting Information) and RSV RNA extracts (**Fig. S16** in revised Supporting Information). Based on the results, we can see the dLAMP can detect ~200 PFU/mL RSV input, similar to that of DIAMOND. Based on the dLAMP results, we estimated that 100 PFU/mL is equivalent to 1 copies/ μ L RNA with our protocols. In other words, DIAMOND has a *LOD* of single-molecule detection, which is competitive with nucleic acid amplification methods. In addition to the comparable sensitivity with digital assays, DIAMOND can detect intact virus without nucleic amplification and additional liquid handling associated with the assay (i.e., virus extraction, thermal incubation, and chip loading). Therefore, we believe DIAMOND has a simplified diagnostic approach and represents an important advance for virus detection. We have added this content in the revised manuscript (Page 8).

Changes on Page 8:

“To compare DIAMOND with other state-of-the-art methods, we performed measurements using digital loop-mediated isothermal amplification (dLAMP). dLAMP is a rapid molecule test and provides absolute quantification of nucleic acids, which has been used to quantify a variety of viruses (33-36). To perform the dLAMP, we used commercially available fluorescent LAMP kit and microwell chips for detecting genomic RNA of RSV (A2 strain) as target. A set of primers (**Table S3**) were used according to a previous publication (37). **Fig. S15a-e** show the fluorescence images of the dLAMP results after incubating at 65 °C for 30 min, where the number of positive wells (brighter) decreases with the concentration of target RNA. A calibration curve ($R^2=0.9998$, **Fig. S15f**) was established as a function of the RNA inputs. The *LOD* was estimated to be 2 copies/ μ L using synthetic RNA (inset of **Fig. S15f**). We then used this method to detect RSV from the spiked samples. The RSV extracts were collected using commercially available RNA extraction buffer and purification kit. **Fig. S16a-d** show the dLAMP detection results for RSV samples at different concentrations. Specifically, RSV of ~200 PFU/mL can be detected (**Fig. S16e**), similar to that of DIAMOND. Furthermore, referring to the calibration curve in **Fig. S15f**, we estimated that 100 PFU/mL is equivalent to 1 copies/ μ L RNA with our protocols (**Fig. S16f**). In other words, DIAMOND has a *LOD* of single-molecule detection, which is competitive with nucleic acid amplification methods.”

Fig. S15. Detection of RSV RNA via digital loop-mediated isothermal amplification (dLAMP). (a-e) Fluorescence images of the microfluidic chips after dLAMP with varied RNA inputs. All images have intensity range of 0-10,000 relative fluorescence units (RFU). (f) A plot of the frequency of positive wells (f_{on}) against different RNA inputs (copies per microliter). The positive wells were determined as the maximum fluorescence intensities larger than a threshold of three standard deviations above the mean. Inset shows the limit of detection (LOD), where the background is set as 3-times standard deviation above the zero calibrator. Each well of the microfluidic chip has a volume of 750 pL.

Fig. S16. Detection of RNA extracts from RSV spiked samples via dLAMP. (a-d) Fluorescence images of the microfluidic chips after dLAMP with varied RSV inputs. All images have intensity range of 0-10,000 RFU. (e) A plot of f_{on} against RSV titers. The background is set as 3-times standard deviation above the zero calibrator. (f) Concentration of RNA detected in RSV extracts based on the background-subtracted frequency (f_{on}') referring to the calibration curve from **Fig. S15f**.

- The technique, demonstrated by the authors is highly interesting for the detection of single viruses, vesicles and cells. However, I fail to see the applicability for the detection of smaller entities such as proteins or DNA molecules. The approach requires clustering of nanoparticles and as such, if a small protein is to be detected, it will most likely be labelled with only one or very few

nanoparticles. According to my understanding, the proposed technique will face its limits for the detection of such molecules and the alternative techniques described in the introduction might be of more use there? A discussion on this matter would be really insightful.

A: We thank the reviewer for pointing out this potential issue. We agree that current DIAMOND system may have difficulty in detecting protein-specific PNB signals from a high background since a protein can be only targeted with a few NPs. As shown in the **Fig. S1** in Supporting Information to Reviewers Only, a potential way to achieve the protein biomarker detection is to detect samples in a diluted setting (e.g., $\lambda = 0.1$, probe per detection zone). This setting is frequently used in the droplet digital ELISA (PNAS, 2019, 116, 4489-4495). Such detection scheme allows discriminating the PNB signals from none, single, and multiple NPs passing through. A discussion on this topic has been added to the revised manuscript (Page 10).

Changes on Page 10:

“It is also possible to explore the feasibility of DIAMOND in detecting analytes that are smaller than viral particles, such as protein biomarkers. This, together with multiplexed detection, is important yet challenging for *in vitro* diagnosis (10, 11, 19). Since only two or several NPs can target on a protein due to its small size and limited binding sites, it is challenging to reliably differentiate protein-specific PNB signals in a high background. Therefore, we can set up a diluted setting with a low NP number per detection zone (e.g., $\lambda = 0.1$, frequently used in droplet digital ELISA) so that the signals from zero, single, and multiple NPs passing through are distinctive (11). Furthermore, it is crucial to ask whether DIAMOND can recognize PNB signals from monomers versus dimers (i.e., two AuNPs in close proximity). As a first step, we have tested and validated a threshold-based gating method (41) to determine the PNB signals generated from a specific number of AuNPs ($k = 0, 1, 2, \dots$, **Fig. S22a**) for the case of $\lambda = 0.4$ (**Fig. 2a**).”

[Redacted]

Fig. S22. Data sorting by a gating method. (a) A bivariate plot of amplitude and AUC extracted from 3,000 pulses for 75 nm AuNPs with $\lambda=0.4$ shown in **Fig. 2a**. The Poisson probabilities are used to determine thresholds ($T=\mu+n \sigma$, μ is mean value, σ is standard deviation, $n>0$) as highlighted by the colors. The frequency (f) of scatters in each color background is counted and shown. Red arrow highlights the scatters with zero amplitude and area-under-curve (AUC). (b) Representative PNB signals of none, single and multiple AuNPs that were extracted from (a) in different color regions. PD is photodetector.

Q. Does the work support the conclusions and claims, or is additional evidence needed?

The authors carried out a wide variety of experiments to characterize the system, substantiated by the extensive body of supplementary information. However, I would like to raise a few questions that should be tackled either in additional discussions or via additional evidence.

- The detection volume is rather big (16 pL), substantiated by the highly relevant comment of the authors about the sampling efficiency. Why did the authors not match the dimensions of the capillary and laser-beam? Are such dimensions not compatible with capillaries? A short note in this respect would be helpful in clarifying this.

Response: We thank the reviewer for bringing up this concern. In the present study, we focused on validating a novel and versatile digital immunoassay. Utilizing a disposable micro-capillary allows us to perform the experiment easily with accessible syringe pump. However, the size mismatch between capillary and laser beam will cause the low sampling efficiency. Therefore, we plan to use a microfluidic system to reduce such mismatch. Using sheath solutions, we can focus the flow and make its diameter close to that of laser beams [redacted]. Optimization on this setup allows reducing the detection volume to ~ 1 pL and thus improving the sampling efficiency. We have added a discussion of the revised manuscript (Page 9).

Changes on Page 9:

“Second, further optimization of the optofluidic system can increase the sampling efficiency for the PNB measurements. In the present system, we used the readily available micro-capillary although only a small portion of the sample is probed by the laser (20-40% along and 5-10% orthogonal to the flow direction, respectively, **Fig. S19**). A low sampling efficiency means less events counting for a given sample volume and thus limits the detection range and sensitivity (5).

A microfluidic flow focusing system can readily solve this problem by creating a narrower flow path (39). Taking advantage of the high-throughput ns laser and focusing flow, we expect to increase the event counting (e.g., 1 million readings within 10 minutes, similar to flow cytometer). This will in turn improve the detection range and sensitivity of DIAMOND (**Fig. S20** and **Table S4**), due to the fact that the digital counting performance essentially relies on the number of events counting (40).”

[Redacted]

- The authors demonstrate that DIAMOND allows size differentiation by showing the efficient discrimination of nanoparticles of 15 nm up to 75 nm particles. It is not clear why the authors chose the 15 nm particles for detection of the SiO₂ beads and the RSV.

Response: We used 15 nm AuNPs as the labels because they can be prepared in relatively uniform in size and shape (**Fig. S2** in revised Supporting Information). Also, we found that the 15 nm AuNPs-based probes have the highest binding efficiency for RSV targeting **[Redacted]**. We have highlighted this in the revised manuscript (Page 7).

Change on Page 7:

“We used 15 nm AuNPs as labels because they can be prepared in relatively uniform in size and shape.”

[Redacted]

- The authors demonstrate that DIAMOND allows identification of heterogeneities in a nanoparticle suspension. However, it is not clear to me why this is relevant in the context of the detection of viral particles. If an RSV particle is labelled with nanoparticles, a cluster of 15 nm nanoparticles arises. This cluster of 15 nm particles should be detected in a background of free-floating 15 nm nanoparticles, rather than a 75 nm nanoparticle in a background of 15 nm particles. Hence the detection of bigger nanoparticles might not be as relevant for the final application as the detection of clusters. It would be interesting to explain the rationale behind this in more detail.

Response: We thank the reviewer for bringing up this question. The rationale is that the clusters may act as bigger particles. So we first tested detecting large 75 nm nanoparticles in the background of small 15 nm nanoparticles. For example, the 75 nm AuNPs can be considered as clusters of 15 nm AuNPs (a 75 nm AuNP = 25 of 15 nm AuNPs in terms of absorption cross-section area). In the manuscript, we use this example (simplest model) to introduce the developed analytical method (threshold and data sorting by code). We have emphasized the rationale this in the revised manuscript (Page 5).

In addition, the ability to identify heterogeneities in a nanoparticle suspension may also allow DIAMOND to be an analytical tool. One of the future applications is to develop DIAMOND for nanoparticle analysis like dynamic light scattering and Exoid (IZON) based on tunable resistive pulse sensing instrument. This is beyond the scope of our current study, and we have briefly discussed this in the revised manuscript (Page 6).

Changes on page 5:

“The rationale is that the NP clusters may act as bigger particles. So we first tested detecting large 75 nm nanoparticles in the background of small 15 nm nanoparticles.”

Changes on page 6:

“This capability may have implications as an analytical tool for NPs and requires further study.”

- The authors refer to their technique as a system that enables fast measurements (within 2 minutes). However, the incubation of RSV with AuNP probes of 30 min does not seem to be included in that time-frame. This is to be clarified as it might be perceived as misleading when comparing DIAMOND to other techniques.

Response: We thank the reviewer for pointing out this issue. We have deleted “fast measurement (within 2 min)” and changed the description in the revised manuscript (Page 4).

Changes on page 4:

“When applied to detect respiratory syncytial virus (RSV), DIAMOND provides good detection specificity among close-related respiratory viruses and achieves a detection limit of ~100 PFU/mL or equivalent to 1 RNA copy/ μ L.”

- It would be highly interesting to include a short discussion on the multiplexing possibilities of this technique, as this is of great importance to the field and thus highly relevant.

Response: We agree with the reviewer that multiplexed detection of biomarkers is an important and relevant aspect for diagnostic applications. As mentioned above, we have discussed the potential of protein detection by DIAMOND. Following that, we can use AuNP probes of different sizes. With the ability to differentiate PNB signals from those probes, DIAMOND may offer a solution to multiplexed detection. **[Redacted]**. A discussion on this topic has been added in the revised manuscript (Page 10).

Changes on Page 10:

“In addition, multiplexed detection is also a critical aspect for a competitive diagnostic method. It may be possible to label the target with AuNP probes that generate different PNB signals. Future studies can focus on demonstrate the feasibility for multiplexing protein detection.”

[Redacted]

- The linear dynamic range depicted in Fig 4e is small and as such, samples in only a very narrow concentration range can be detected reliably. Is this a given limitation of the system or is there a workaround for the future? If not, how do the authors foresee to deal with that for testing of real samples?

Response: We thank the reviewer for pointing out this issue. Here the linear dynamic range is about 2 logs and the total dynamic range is about 4 logs ($\lambda=0.001$ to 10 or 100 aM-0.1 pM). The short linear range is essentially limited by the counting number (i.e., 3,000 pulses). There are two ways to extend the dynamic range. First, increasing nanobubble counts can improve the dynamic range as routinely done in digital assays in order to detect rare targets, for example from 10^4 to 10^6 or 10^7 (Lab Chip 2012, 12, 4986-4991). Second, analog detection can avoid saturating the digital counts at high concentrations (*ACS Nano* 2018, 12, 5880-5887) as the sensitivity is no longer a limitation in this range. It is entirely possible to combine the colorimetric analog detection with the current digital nanobubble detection for detecting analytes at a wider dynamic range.

We have highlighted this in the revised manuscript (Page 6 and 9).

Changes on page 6:

“It should be pointed out that the detection range of DIAMOND only covers 2 logs due to the limited counting number (i.e., 3,000 pulses). Alternatively, we can use an analogy method for the analysis of PNB signals (e.g., averaged AUC versus analyte concentration), which should provide additional detection range beyond 100% f_{on} (13).”

Changes on page 9:

“Taking advantage of the high-throughput ns laser and focusing flow, we expect to increase the event counting (e.g., 1 million readings within 10 minutes, similar to flow cytometer). This will in turn improve the detection range and sensitivity of DIAMOND (Fig. S20 and Table S4), due to the

fact that the digital counting performance essentially relies on the number of events counting (40).”

- Figure 3b: Please indicate i, ii and iii. Also, the caption described a red arrow but the arrow is grey.

Response: As suggested, we have added the description of *i-iii* and corrected the typo.

Q. Are there any flaws in the data analysis, interpretation and conclusions? - Do these prohibit publication or require revision?

- Figure 4b: Please indicate where the RSV is. If I understand it correctly, I see clusters of nanoparticles aside of the RSV (on Fig 4b(i) at right bottom side, in 4b(ii) at top center?). How will these clusters aside of the RSV reflect in a signal in the DIAMOND? Will these clusters remain attached to the RSV or will they result in free-floating clusters and thus additional signals? Overall, it is not clear to me how the authors prevent or anticipate aggregation of NPs in the sample and a brief discussion on this topic is needed.

Response: We thank the reviewer for pointing out this issue. In **Fig. 5b**, we have highlighted the boundaries of the RSV in red to make it clear. Those aggregated nanoparticles aside of the RSV should also be attached to the RSV, instead of free-floating. This may be due to the staining issue that did not fully resolve the RSV in TEM image.

To reduce non-specific binding of NP probes, we further explored backfilling of antibody-conjugated nanoparticles with BSA. The results suggest that this strategy is highly effective (new **Fig. 6**). To further quality check any non-specific binding of NP probes, a good practice such as running a reference sample as negative control should allow us to remove the background signals. Our future optimization of DIAMOND will work on laser modulation, in order to apply low-fluence pulse laser that only activate large clusters for PNB generation. We have added this in the discussion part of the revised manuscript (Page 9-10).

Changes on page 9-10:

“Lastly, it is worth exploring different modes of DIAMOND operation. Since the PNB generation is dependent on the laser fluence, we can perform the DIAMOND test in two modes by modulating the laser fluence above and below PNB generation threshold, referred to as above- and below-threshold modes (21). Currently we used the above-threshold mode with high laser fluence at $P_{\text{PNB}}=100\%$ (blue arrow of **Fig. S21**), resulting in the generation of PNB signals from both the small NPs and larger clusters. Alternatively, we may adopt the below-threshold mode at a lower laser fluence at $P_{\text{PNB}}=0\%$ (red arrow of **Fig. S21**) to only activate large clusters for PNB generation.

Q. Is the methodology sound? Does the work meet the expected standards in your field?

- The authors demonstrate the use of DIAMOND in transporting medium. However, the performance of the system in nasal swabs of healthy controls, spiked with RSV is required to enable proper assessment of the proposed technique and to meet the standards in the field.

A: We thank the reviewer for this suggestion. Yes, we agree with the reviewer that testing samples in complex matrix is important to clarify the DIAMOND's application of virus detection. We have tested virus sample spiked in nasal swabs of healthy controls. Indeed, we found that the original probes did have non-specific binding issues. To address this issue, we backfilled bovine serum albumin (BSA) on the AuNP probes. The results (new **Fig. 6** of the revised manuscript, see below) indicate that DIAMOND can selectively differentiate respiratory syncytial virus among other close-related viruses and achieve a detection limit of 102 PFU/mL. This result suggests that with BSA-backfilled probes, DIAMOND can work for virus in a complex matrix, which is promising for clinical sample test. We have added this content in the revised manuscript (Page 7-8).

Changes on Page 7-8:

“To further demonstrate the potential clinical applications, we applied DIAMOND to detect RSV spiked in the nasal swab samples. Viral transportation medium (VTM), hMPV, PIV, and IVA were used as negative controls and all viral titers were kept same as 5×10^4 PFU/mL. We first used the Synagis-labeled AuNPs as probes for the virus detection (inset of **Fig. 6a**). The detection results (**Fig. S11** and **Fig. 6a, b**) suggest positive PNB signals (above the thresholds) for all control viruses (hMPV, PIV, and IVA) using RSV-specific AuNP probes. This phenomenon can be ascribed to the non-specific binding between the probes and impurities (e.g., cell debris) included in the unpurified nasal swab samples. To address this issue, we used bovine serum albumin (BSA) to backfill the AuNP probes (inset of **Fig. 6c** and **Fig. S12**). The BSA as block reagents have been frequently used in immunoassays and can prevent the non-specific binding for improved detection specificity (32). **Fig. S13** and **Fig. 6c, d** show the virus detection results using BSA-backfilled probes, where the positive PNB signals from the control viruses were significantly reduced. In contrast, the PNB signals from RSV can be easily distinguished from the control samples, yielding $f_{on}=100\%$ that matches well with the theoretical probability. This result suggests a better detection specificity of DIAMOND utilizing BSA-backfilled probes. Similarly, these probes enable sensitive detection of spiked RSV nasal swab samples with detection limit of 102 PFU/mL (**Fig. S14** and **Fig. 6e, f**), similar to purified viruses (**Fig. 5**). Taken together, these data demonstrated that DIAMOND implementing on homogeneous immunoassay allows sensitive analysis of viral samples in complex matrix and supports the potential clinical applications.”

Fig. 6. Detection of respiratory viruses spiked in nasal swab samples by DIAMOND. (a, c, e) Bivariate plots of amplitude and AUC extracted from 3,000 pulses for the assay solutions incubating different respiratory viruses with (a) AuNP-Synagis as probes, (c) bovine serum albumin (BSA)-backfilled AuNP-Synagis probes, and (e) RSV of different titers (PFU/mL) with BSA-backfilled AuNP-Synagis probes. (b, d, f) Corresponding f_{on} counted from (a, c, e) against (b, d) different respiratory viruses and (f) RSV titers, respectively. VTM is viral transportation medium. Insets in (a, c, e) show the model of AuNP-based probes used correspondingly. Inset in (f) shows the linear detection range and as-determined LOD .

Q. Is there enough detail provided in the methods for the work to be reproduced?

- The methods are clear and include sufficient detail to reproduce the work.

Response: We thank the reviewer for all the comments that help improve the quality of our manuscript.

Reviewer #3 (Remarks to the Author):

The authors report on a particle-based biosensing assay that uses bubbles generated by photothermal heating of nanoparticle assemblies. The presence of analyte leads to assembly of plasmonic particles on a scaffold, e.g. a virus particle. The plasmonic assemblies are flown through a fluidic channel and illuminated by a pulsed laser, causing nanobubble generation. This provides a non-linear readout that is strongly biased to the assemblies allowing for the counting of the number of assemblies against a background of free particles in solution. The authors show several benchmarking experiments meant to optimize the particle size and concentration, and finally show the application to a single-step immunoassay for virus detection. I think the assay format is well presented and tested. I believe the manuscript would profit from additional measurements and a fairer comparison to the state-of-the-art to allow for a better description of the assay performance.

Response: We thank the reviewer for the encouraging comments and have addressed each comment below.

In particular:

- although the assay is new as far as I know, the manuscript contains a rather lengthy methodological description with little actual sensing data. It would have been nice to show the ability to detect multiple analytes, but I can imagine this is not an easy feat at this stage of the technology. Importantly though, controls are shown in figure 5 for colorimetric detection, but the controls in the nanobubble assay are not mentioned. The controls should be performed and shown in Figure 5e for non-complementary gold particles or different virus types to demonstrate the specificity of the detected signals.

Response: We thank the reviewer for this suggestion. The reviewer is correct that detecting different analytes is early at this stage of technology development. For other analytes such as protein detection, the challenge is to label target protein with at least two nanoparticles to generate a significantly different signal. We are actively looking into methods for protein biomarker detection as a next step of the technology development (see below). In this manuscript, we also have made an important step forward by proving a sensitive detection using virus particles in a complex media that is equivalent to single copy RNA detection. In terms of the control virus detection, we have added additional experiments and controls in new **Fig. 6** of revised manuscript.

In terms of protein biomarker detection, a potential way is to detect samples in a diluted setting (e.g., $\lambda = 0.1$, probe per detection zone). See schematics in **Fig. S1** in Supporting Information to Reviewers Only. This setting is frequently used in the droplet digital ELISA (PNAS, 2019, 116, 4489-4495). Such detection scheme allows discriminating the PNB signals from none, single, and multiple NPs passing through. A discussion on this topic has been added to the revised manuscript (Page 10).

Changes on Page 10:

“It is also possible to explore the feasibility of DIAMOND in detecting analytes that are smaller than viral particles, such as protein biomarkers. This, together with multiplexed detection, is important yet challenging for *in vitro* diagnosis (10, 11, 19). Since only two or several NPs can target on a protein due to its small size and limited binding sites, it is challenging to reliably differentiate protein-specific PNB signals in a high background. Therefore, we can set up a diluted setting with a low NP number per detection zone (e.g., $\lambda = 0.1$, frequently used in droplet digital ELISA) so that the signals from zero, single, and multiple NPs passing through are distinctive (11). Furthermore, it is crucial to ask whether DIAMOND can recognize PNB signals from monomers versus dimers (i.e., two AuNPs in close proximity). As a first step, we have tested and validated a threshold-based gating method (41) to determine the PNB signals generated from a specific number of AuNPs ($k = 0, 1, 2, \dots$, Fig. S22a) for the case of $\lambda = 0.4$ (Fig. 2a).”

[Redacted]

Fig. S22. Data sorting by a gating method. (a) A bivariate plot of amplitude and AUC extracted from 3,000 pulses for 75 nm AuNPs with $\lambda=0.4$ shown in Fig. 2a. The Poisson probabilities are used to determine thresholds ($T=\mu+n\sigma$, μ is mean value, σ is standard deviation, $n>0$) as highlighted by the colors. The frequency (f) of scatters in each color background is counted and shown. Red arrow highlights the scatters with zero amplitude and area-under-curve (AUC). (b) Representative PNB signals of none, single and multiple AuNPs that were extracted from (a) in different color regions. PD is photodetector.

- the sensitivity of the assay is now compared to lateral flow and colorimetric assays, which are indeed simple but do not provide state-of-the-art sensitivity. A fairer comparison in terms of sensitivity would be to involve commercially available digital assays (e.g. ELISA, PCR, and others, see e.g. Lab Chip 2020, 20, 2816-2840). Particularly the comparison to flow cytometry approaches is relevant as the setup design is similar. These commercially available approaches are now mentioned in passing in the introduction and in the supplementary material, but a quantitative comparison is needed to judge the benefits of the presented method.

Response: We thank the reviewer for bringing up this concern. We agree with the reviewer that a comparison between DIAMOND and other digital assay is needed to clarify the advantages. We have conducted the digital loop-mediate isothermal amplification (dLAMP) for comparison. dLAMP is a rapid molecule test and detects the amplified DNA via fluorescent microscope. We used commercially available LAMP kit and microwell chip for the reaction and designed a MATLAB code for data analysis. Below figures are the dLAMP detection results of synthetic RNA (**Fig. S15** in revised Supporting Information) and RSV RNA extracts (**Fig. S16** in revised Supporting Information). Based on the results, we can see the dLAMP can detect ~200 PFU/mL RSV input, similar to that of DIAMOND. Based on the dLAMP results, we estimated that 100 PFU/mL is equivalent to 1 copies/ μ L RNA with our protocols. In other words, DIAMOND has a LOD of single-molecule detection, which is competitive with nucleic acid amplification methods. In addition to the comparable sensitivity with digital assays, DIAMOND can detect intact virus without nucleic amplification and additional liquid handling associated with the assay (i.e., virus extraction, thermal incubation, and chip loading). Therefore, we believe DIAMOND has a simplified diagnostic approach and represents an important advance for virus detection. We have added this content in the revised manuscript (Page 8).

Changes on Page 8:

“To compare DIAMOND with other state-of-the-art methods, we performed measurements using digital loop-mediated isothermal amplification (dLAMP). dLAMP is a rapid molecule test and provides absolute quantification of nucleic acids, which has been used to quantify a variety of viruses (33-36). To perform the dLAMP, we used commercially available fluorescent LAMP kit and microwell chips for detecting genomic RNA of RSV (A2 strain) as target. A set of primers (**Table S3**) were used according to a previous publication (37). **Fig. S15a-e** show the fluorescence images of the dLAMP results after incubating at 65 °C for 30 min, where the number of positive wells (brighter) decreases with the concentration of target RNA. A calibration curve ($R^2=0.9998$, **Fig. S15f**) was established as a function of the RNA inputs. The *LOD* was estimated to be 2 copies/ μ L using synthetic RNA (inset of **Fig. S15f**). We then used this method to detect RSV from the spiked samples. The RSV extracts were collected using commercially available RNA extraction buffer and purification kit. **Fig. S16a-d** show the dLAMP detection results for RSV samples at different concentrations. Specifically, RSV of ~200 PFU/mL can be detected (**Fig. S16e**), similar to that of DIAMOND. Furthermore, referring to the calibration curve in **Fig. S15f**, we estimated that 100 PFU/mL is equivalent to 1 copies/ μ L RNA with our protocols (**Fig. S16f**). In

other words, DIAMOND has a LOD of single-molecule detection, which is competitive with nucleic acid amplification methods.”

Fig. S15. Detection of RSV RNA via digital loop-mediated isothermal amplification (dLAMP). (a-e) Fluorescence images of the microwell chips after dLAMP with varied RNA inputs. All images have intensity range of 0-10,000 relative fluorescence units (RFU). (f) A plot of the frequency of positive wells (f_{on}) against different RNA inputs (copies per microliter). The positive wells were determined as the maximum fluorescence intensities larger than a threshold of three standard deviations above the mean. Inset shows the limit of detection (LOD), where the background is set as 3-times standard deviation above the zero calibrator. Each well of the microfluidic chip has a volume of 750 pL.

Fig. S16. Detection of RNA extracts from RSV spiked samples via dLAMP. (a-d) Fluorescence images of the microwell chips after dLAMP with varied RSV inputs. All images have intensity range of 0-10,000 RFU. (e) A plot of f_{on} against RSV titers. The background is set as 3-times standard deviation above the zero calibrator. (f) Concentration of RNA detected in RSV extracts based on the background-subtracted frequency (f_{on}') referring to the calibration curve from **Fig. S15f**.

- the authors claim in the introduction that other digital assays are "complex". It is not specified what the complexity is they refer to, and in what sense the current assay is easier.

Response: We thank the reviewer for pointing out this issue. The complexity of digital immunoassays refers to the assay operation, such as the necessities of washings, sample partitioning, and signal amplification prior to the measurements. In contrast, DIAMOND works on

a homogeneous immunoassay that bypasses the need for the liquid handling and is easy-to-use. We have explained the complex of current digital immunoassays in the revised manuscript (Page 3).

Changes on Page 3:

“Despite these advantages, digital immunoassays have complex assay protocols, such as multiple washing steps, sample partitioning, and signal amplification prior to the measurements, that have limited its widespread use.”

- The discussion now heavily focuses on the pulse frequency to increase the speed of the assay. It would be nice to increase the scope of the discussion. For example: the presented assay requires a pulsed and a CW laser that are overlapped in their focus. This is not a cheap setup and not easily achieved in other labs. Can the authors discuss future improvements needed to bring the assay to a broad range of labs? Also, is the method easily extended to other analytes, and if so, which ones are most obvious candidates?

Response: As suggested, we have discussed the potential optimizations in order to bring this technique to a broad range of labs and practical applications. In one direction, we aim to use small nanosecond laser, microfluidic system, and different operation mode (low laser fluence) that allow us to develop benchtop device, more efficient events counting, better detection performance, and simplified operation. In the other direction, we aim to apply DIAMOND for protein biomarkers and multiplex analytes, which is important yet challenging for *in vitro* diagnosis. We have highlighted those discussions in the revised manuscript (Page 8-10).

Current implementation of DIAMOND is particular suitable for large biological particles with abundant binding sites (accumulate as many probes as possible). Examples include extracellular vesicles (ectosomes, exosomes) that are promising biomarkers for cancers.

Changes on Page 8-10:

“Digital immunoassays create high standards as next-generation diagnostic platforms, such as calibration-free quantification and single-molecule detection of biomarkers for early diagnostics. The major barriers for its widespread use are the time-consuming protocols and laboratory infrastructures. Here we developed DIAMOND to overcome some of these bottlenecks. Specifically, DIAMOND uses a homogeneous immunoassay format and does not require additional sample washing, separation, and signal amplification steps (4, 6). On the other hand, compared to the digital homogeneous assays, DIAMOND does not require chip preparation and on-chip reaction and can be performed with less assay time (18-20). Also, DIAMOND can detect intact virus without additional liquid handling (i.e., virus extraction, thermal incubation, and chip loading), offering a simplified diagnostic approach. Therefore, we believe DIAMOND is highly promising as a simple and ultrasensitive digital diagnostic platform.

In the present study, we focused on developing and validating a novel and versatile digital immunoassay. We envision several further improvements to this technology in order to bring it to

a broad range of labs and practical applications. First, it is possible to design a small benchtop device for the plasmonic nanobubble (PNB) measurements that can be distributed to other labs (38). By replacing the research-grade picosecond pulse laser with a smaller nanosecond (ns) laser (e.g., Wedge-HB-532, RPMC), all components can be integrated in a benchtop device (15×15×6 inches, **Fig. S17**). Evaluation of PNB generation and detection by this ns laser shows robust results across a range of AuNP concentrations (**Fig. S18**). More importantly, the ns laser provides a repetition rate up to 2,000 Hz and allows a much faster readout and more efficient event counting. Second, further optimization of the optofluidic system can increase the sampling efficiency for the PNB measurements. In the present system, we used the readily available micro-capillary although only a small portion of the sample is probed by the laser (20-40% along and 5-10% orthogonal to the flow direction, respectively, **Fig. S19**). A low sampling efficiency means less events counting for a given sample volume and thus limits the detection range and sensitivity (5). A microfluidic flow focusing system can readily solve this problem by creating a narrower flow path (39). Taking advantage of the high-throughput ns laser and focusing flow, we expect to increase the event counting (e.g., 1 million readings within 10 minutes). This will in turn improve the detection range and sensitivity of DIAMOND (**Fig. S20** and **Table S4**), due to the fact that the digital counting performance essentially relies on the number of events counting (40). Lastly, it is worth exploring different modes of DIAMOND operation. Since the PNB generation is dependent on the laser fluence, we can perform the DIAMOND test in two modes by modulating the laser fluence above and below PNB generation threshold, referred to as above- and below-threshold modes (21). Currently we used the above-threshold mode with high laser fluence at $P_{\text{PNB}}=100\%$ (blue arrow of **Fig. S21**), resulting in the generation of PNB signals from both the small NPs and larger clusters. Alternatively, we may adopt the below-threshold mode at a lower laser fluence at $P_{\text{PNB}}=0\%$ (red arrow of **Fig. S21**) to only activate large clusters for PNB generation.

It is also possible to explore the feasibility of DIAMOND in detecting analytes that are smaller than viral particles, such as protein biomarkers. This, together with multiplexed detection, is important yet challenging for *in vitro* diagnosis (10, 11, 19). Since only two or several NPs can target on a protein due to its small size and limited binding sites, it is challenging to reliably differentiate protein-specific PNB signals in a high background. Therefore, we can set up a diluted setting with a low NP number per detection zone (e.g., $\lambda = 0.1$, frequently used in droplet digital ELISA) so that the signals from zero, single, and multiple NPs passing through are distinctive (11). Furthermore, it is crucial to ask whether DIAMOND can recognize PNB signals from monomers versus dimers (i.e., two AuNPs in close proximity). As a first step, we have tested and validated a threshold-based gating method (41) to determine the PNB signals generated from a specific number of AuNPs ($k = 0, 1, 2, \dots$, **Fig. S22a**) for the case of $\lambda = 0.4$ (**Fig. 2a**). In addition, multiplexed detection is also a critical aspect for a competitive diagnostic method. It may be possible to label the target with AuNP probes that generate different PNB signals. Future studies can focus on demonstrate the feasibility for multiplexing protein detection.”

minor comments:

- on page 4 on line 23 it is stated that the theoretical probability is predicted by Poisson statistics but it is not explained how.

Response: As suggested, we have included the Poisson statistics equation for illustration of how the theoretical probability is calculated.

- on page 5 at the top it is stated that the distribution profiles could be fitted with 1.98 and 3.51-order dependencies. Why is this indicative of a mechanism involving nanobubbles?

Response: We thank the reviewer for bringing up this question. To our knowledge, the generation of nanobubbles is a photothermal response of AuNP under laser excitation. Since the PNB is the amplitude and lifetime values of PNB signal are proportional to the absorption cross-section area of AuNPs. The width (lifetime) or height (amplitude) of a PNB signal is proportional to 2-order to the NPs' size. While the AUC is an integral area of a PNB peak (integration of amplitude along lifetime), it is expected to be roughly 4-order to the NPs' size. In our study, the amplitude and AUC values were fitted with 1.98 and 3.51-order dependencies, close to the theoretical values. The variation of AUC value may come from the filtering of raw data prior to the peak analysis.

- in figures 2b/2c/4e/4f/5c/5e: what are the red lines?

Response: In Figure 2b/4e/5c/5e, the red lines are logistic fitting curves and in Figure 2c/4e inset/4f/5e inset, the red lines are linear fitting of the data. All the fitting was conducted in the Origin software. We have added the description for each image.

- in Fig 5b it is unclear to me what we observe here, I see fibrous material with gold particles on top, is that expected?

Response: The TEM images shown in **Fig. 5b** are AuNPs targeting on RSV. The RSV have low contrast in the images. To make it clear, we have highlighted the boundaries of the RSV in red as below.

Fig. 5. Detection of respiratory syncytial virus (RSV) in a one-step homogeneous immunoassay by DIAMOND. (a) The schematic illustration of a homogeneous immunoassay for RSV utilizing antibody-conjugated AuNPs as probes at room temperature (RT). (b) TEM images of AuNP probes targeting RSV. The boundaries of RSV were highlighted in red. (c) Colorimetric analysis of the AuNP-based homogeneous immunoassay for different respiratory viruses. hMPV is Human metapneumovirus, PIV is Parainfluenza viruses, and IVA is Influenza A. (d) Bivariate plot of amplitude and AUC extracted from 3,000 pulses for the assay solutions with RSV at different titers (5×10^4 PFU/mL). Dashed lines indicate the positions of thresholds calculated from the control sample. (e) Quantification of RSV titers as a function of f_{on} . Inset shows the linear detection range. Error bars in (c) and (e) indicate the standard deviations of three independent measurements, the data was fitted by logistic fitting, and the LOD was calculated as 3 standard deviations of the control divided by the slope of regression line.

- the main text discusses little about the fluidics, e.g. the design of the microfluidic channels, the flow speed, spot size etc. It would be nice to move certain aspects from the SI to the main text to better describe the method.

Response: As suggested, we have added detail description in the revised manuscript (Page 4). Also, we have discussed the potential method to address the size mismatch between the laser beams and micro-capillary in Page 9 of the revised manuscript. [Redacted].

Changes on Page 4:

“Serial aqueous dilutions of 75 nm AuNPs (as characterized in **Fig. S2, 3** and **Table S2**) flowed through a 200 μm micro-capillary using a syringe pump at a speed of 6 $\mu\text{L}/\text{min}$ and irradiated by the pump laser with repetition rate of 50 Hz and beam size of 10 μm (**Fig. S1**). Such configuration allows a laser scanning speed of 1,000 $\mu\text{m}/\text{s}$ and flow speed of 2,500 $\mu\text{m}/\text{s}$, ensuring PNB detection without overlapping.”

Changes on Page 9:

“A microfluidic flow focusing system can readily solve this problem by creating a narrower flow path (39).”

[Redacted]

We thank the reviewer for all the comments that help improve the quality of our manuscript.

REVIEWERS' COMMENTS

Reviewer #1 (Remarks to the Author):

I thank the authors for carrying out the additional work to address my initial comments. They have done a very good job of responding to my concerns.

A few additional comments and questions:

1. The first two pages of the Supplementary materials file is cut off on the right-hand side
2. I am not sure I understand why the signal intensities in Figure 2a decrease at the two lowest concentrations $\lambda=0.4$ and 0.04 . At single particle detection, the signal amplitudes should be similar.
3. Although the authors demonstrate single particle sensitivity, the new work with spiked in virus shows many nanoparticles attached to each viral particle. With single nanoparticle sensitivity, I don't understand how free PNPs are distinguished from virus.
4. It looks like aggregation occurs with the labeled virus.

Reviewer #2 (Remarks to the Author):

I am pleased to see the authors took all feedback into consideration, made the required adjustments, provided additional data to support the results and included the suggested sections in the discussion. As such, they drastically improved the quality of the manuscript. I do not have remaining concerns or points of attention that were not tackled by the authors in the previous revision round.

Reviewer #3 (Remarks to the Author):

The authors have largely answered the questions raised by the referees. One remaining question, which was not answered by the authors, considers the controls shown in Figure 5. The controls are now only shown for a different assay (colorimetric), whereas controls for the bubble assay are crucial for the correct interpretation of results. I am sure the authors have done these controls, so once they are integrated in the manuscript I believe it is ready for publication.

Response Letter to Reviewers

Reviewer #1 (Remarks to the Author):

I thank the authors for carrying out the additional work to address my initial comments. They have done a very good job of responding to my concerns.

A few additional comments and questions:

1. The first two pages of the Supplementary materials file are cut off on the right-hand side.

Response: We thank the reviewer for carefully reading our work. We have generated a PDF file for Supplementary materials without cut-off.

2. I am not sure I understand why the signal intensities in Figure 2a decrease at the two lowest concentrations $\lambda=0.4$ and 0.04 . At single particle detection, the signal amplitudes should be similar.

Response: We thank the reviewer for raising this point. In the case of $\lambda=0.4$, there will be a few NPs passing through together (see **Fig. S4** for Poisson statistics and **S22** for experimental observation); while in the case of 0.04 , only single NP will pass through. This makes the signal different.

Fig. S4. Probability distribution histograms of the AuNPs number (k) in each virtual compartment for a given λ , based on Poisson statistics.

Fig. S22. Data sorting by a gating method. (a) A bivariate plot of amplitude and AUC extracted from 3,000 pulses for 75 nm AuNPs with $\lambda=0.4$ shown in **Fig. 2a**. The Poisson probabilities are used to determine thresholds ($T=\mu+n\sigma$, μ is mean value, σ is standard deviation, $n>0$) as highlighted by the colors. The frequency (f) of scatters in each color background is counted and shown. Red arrow highlights the scatters with zero amplitude and area-under-curve (AUC). (b) Representative PNB signals of none, single and multiple AuNPs that were extracted from (a) in different color regions. PD is photodetector.

3. Although the authors demonstrate single particle sensitivity, the new work with spiked in virus shows many nanoparticles attached to each viral particle. With single nanoparticle sensitivity, I don't understand how free PNPs are distinguished from virus.

Response: We thank the reviewer for pointing out this issue. The detection principles of nanoparticles and viruses are indeed different. For nanoparticle detection, DIAMOND can detect nanobubble signals generated from a single particle flowing through (e.g., $\lambda=0.04$, **Fig. 2a-c**). That's how we demonstrate single nanoparticle sensitivity. While for virus detection, DIAMOND detects nanobubble signals generated from free AuNP probes and virus-AuNPs complex. Because the signal from the virus-AuNPs complex is much stronger than that of free probes, we can differentiate it in a digital manner (“on” and “off”) and thus quantify the virus concentration using Poisson statistics. To make it clear we added a scheme in **Fig. 1c** to illustrate the virus detection mechanism.

Fig. 1. Concept of Digital plasMONic nanobubble Detection (DIAMOND). (a) The schematic illustration of the plasmonic nanobubbles (PNBs) generation and detection in flow. The gold nanoparticles (AuNPs) as labels are used for the generation of the PNBs by short laser pulses and subsequently detected by a secondary probe laser due to the amplified optical absorption. (b, c) The principle of digital counting for NP (b) and virus (c) detection. The “on” and “off” in (b) and (c) refer to the positive and negative PNB signals representing for the presence or absence of targets (i.e., NP and virus, respectively). The laser pulses create “virtual” detection zones as compartmentations for digital counting of PNB signals.

4. It looks like aggregation occurs with the labeled virus.

Response: We thank the reviewer for bringing up this issue. To image viruses by transmission electron microscopy (TEM), we stained the viral particles at a high concentration (10^6 PFU/mL). This may cause the viruses aggregating. It is also possible that the staining and drying process of the sample solution on the TEM grid causes the aggregation.

We thank the reviewer again for all the comments that greatly improve the quality of our manuscript.

Reviewer #2 (Remarks to the Author):

I am pleased to see the authors took all feedback into consideration, made the required adjustments, provided additional data to support the results and included the suggested sections in the discussion. As such, they drastically improved the quality of the manuscript. I do not have remaining concerns or points of attention that were not tackled by the authors in the previous revision round.

Response: We thank the reviewer for examining our manuscript.

Reviewer #3 (Remarks to the Author):

The authors have largely answered the questions raised by the referees. One remaining question, which was not answered by the authors, considers the controls shown in Figure 5. The controls are now only shown for a different assay (colorimetric), whereas controls for the bubble assay are crucial for the correct interpretation of results. I am sure the authors have done these controls, so once they are integrated in the manuscript I believe it is ready for publication.

Response: We thank the reviewer for raising this point. We have performed the DIAMOND test for those control viruses dispersed in 2 mM borate buffer and provided the results in **Fig. S11** and **Fig. 6a, b**. Using non-BSA-backfilled AuNP probes, we observed positive PNB signals for the controls that were probably resulted from the non-specific aggregation of probes. Because those control viruses were received in cell culture fluid that contains impurities like cell debris and used without purification, it leads to the non-specific binding. The nonspecific binding problem was solved by using BSA-backfilling on AuNP probes (**Fig. 6c-f**). We have discussed this in the revised manuscript.

Changes on Page 7:

“It should be mentioned that when used for control virus detection (i.e., hMPV, PIV, and IVA, dispersed in borate buffer), the RSV-specific AuNP probes caused non-specific binding as suggested by the positive PNB signals (**Fig. S11** and **Fig. 6a, b**). This phenomenon can be ascribed to that the control viruses were received in cell culture fluid that contains impurities like cell debris and thus leads to the non-specific aggregation.”

Fig. S11. Representative PNB signal traces (100 pulses) for the assay solutions that incubating Au-Synagis probes with suspensions of different respiratory viruses. The Ctrl is 2 mM borate buffer used as a control. hMPV is Human metapneumovirus, PIV is Parainfluenza viruses, and IVA is Influenza viruses A. Those control viruses are used as received without purification. The concentration of all viruses was kept the same as 5×10^5 PFU/mL in borate buffer.

Note: The as-received control viruses are dispersed in cell culture fluid that contains impurities like additives and cell debris that may cause the non-specific aggregation of Au probes.

Fig. 6. Detection of closely related respiratory viruses by DIAMOND. (a, c, e) Bivariate plots of the amplitude and AUC extracted from 3,000 pulses for the assay solutions incubating (a) AuNP-Synagis as probes with different respiratory viruses in borate buffer, (c) bovine serum albumin (BSA)-backfilled AuNP-Synagis probes with different respiratory viruses spiked in the nasal swab samples, and (e) BSA-AuNP probes with RSV of different titers (PFU/mL) spiked in the nasal swab samples. (b, d, f) The corresponding f_{on} counted from (a, c, e) against (b, d) different respiratory viruses and (f) RSV titers, respectively. VTM is viral transport medium. Insets in (a, c, e) show the models of AuNP-based probes used correspondingly. Inset in (f) shows the linear detection range and as-determined LOD. Error bars in (b), (d), and (f) indicate the standard deviations of three independent measurements.